# Probing the structure of water in individual living cells

Xiaoqi Lang[1], Lixue Shi [2], Zhilun Zhao [1] & Wei Min [1] ✉

Water regulates or even governs a wide range of biological processes. Despite its fundamental importance, surprisingly little is known about the structure of intracellular water. Herein we employ a Raman micro-spectroscopy technique to uncover the composition, abundance and vibrational spectra of intracellular water in individual living cells. In three different cell types, we show a small but consistent population (~3%) of non-bulk-like water. It exhibits a weakened hydrogen-bonded network and a more disordered tetrahedral structure. We attribute this population to biointerfacial water located in the vicinity of bio-molecules. Moreover, our whole-cell modeling suggests that all soluble (globular) proteins inside cells are surrounded by, on average, one full molecular layer (about 2.6 Angstrom) of biointerfacial water. Furthermore, relative invariance of biointerfacial water is observed among different single cells. Overall, our study not only opens up experimental possibilities of interrogating water structure in vivo but also provides insights into water in life.

Water, the active matrix of life, is an integral part of the structural organization of living cells[1]. It is ubiquitous for the existence of all known life forms. Water mediates or even governs many, if not all, vital biological interactions inside of a cell, such as protein folding, enzyme catalysis, membrane self-assembly, and substrate recognition[2–6]. Despite the importance, the structure of intracellular water has remained elusive[7], although the dynamics of a portion of intracellular water was found to be slowed down[8–18]. After all, liquid water owes most of its physicochemical properties to the structure of the hydrogen-bonding network, which provides the underlying framework for understanding the structure-dynamics-function relationship. Fundamental questions regarding intracellular water are either not addressed or remain highly debated: whether the hydrogen-bonding network of water inside living cells differs from that of bulk water (in fact, one school of thought suggests little to no bulk-like intracellular water[19–25]), and if so, what are the abundance and location of such water, and what are the new structural features (such as tetrahedrality, hydrogen-bonding strength and dangling bonds) of this "non-bulk-like" water?

The lack of understanding of intracellular water is largely due to experimental difficulties in probing the structure of water's hydrogen-bonding network in living cells. To this end, vibrational spectroscopy provides a promising approach, as the O-H stretching vibration carries valuable information about the local hydrogen-bonding structure[26–29]. However, living cells present a formidable challenge. Vibrational sum frequency generation (SFG) has intrinsic interface selectivity but requires an extended planar interface[30,31], which is not compatible with intracellular water. Chiral SFG, leveraging chirality transfer from bio-molecules to adjacent water, allows probing of interfacial water in vitro without the necessity for a flat surface[32,33]. Vibrational sum frequency scattering can probe the surface of submicron particles in suspension but cannot study biomolecules (such as proteins) that are much smaller than the wavelength of light[34,35]. Unlike the second-order optical processes, infrared absorption and Raman scattering could directly measure the vibrational spectrum of water inside living cells. However, because they are probing volume properties (i.e., lack of interface selectivity), it is deemed difficult to distinguish bulk-like water and water in the vicinity of biomolecules (which we name "biointerfacial water"), with the former likely dominating over the latter. Moreover, they have to face the complication of interfering solutes inside live cells: N-H bonds, mainly from protein backbone and side chains, unavoidably contribute to the vibrational signal in the O-H

[1]Department of Chemistry, Columbia University, New York, NY 10027, USA. [2]Shanghai Xuhui Central Hospital, Zhongshan-Xuhui Hospital, and Shanghai Key Laboratory of Medical Epigenetics, International Co-laboratory of Medical Epigenetics and Metabolism, Institutes of Biomedical Sciences, Shanghai Medical College, Fudan University, Shanghai 200032, China. ✉e-mail: wm2256@columbia.edu

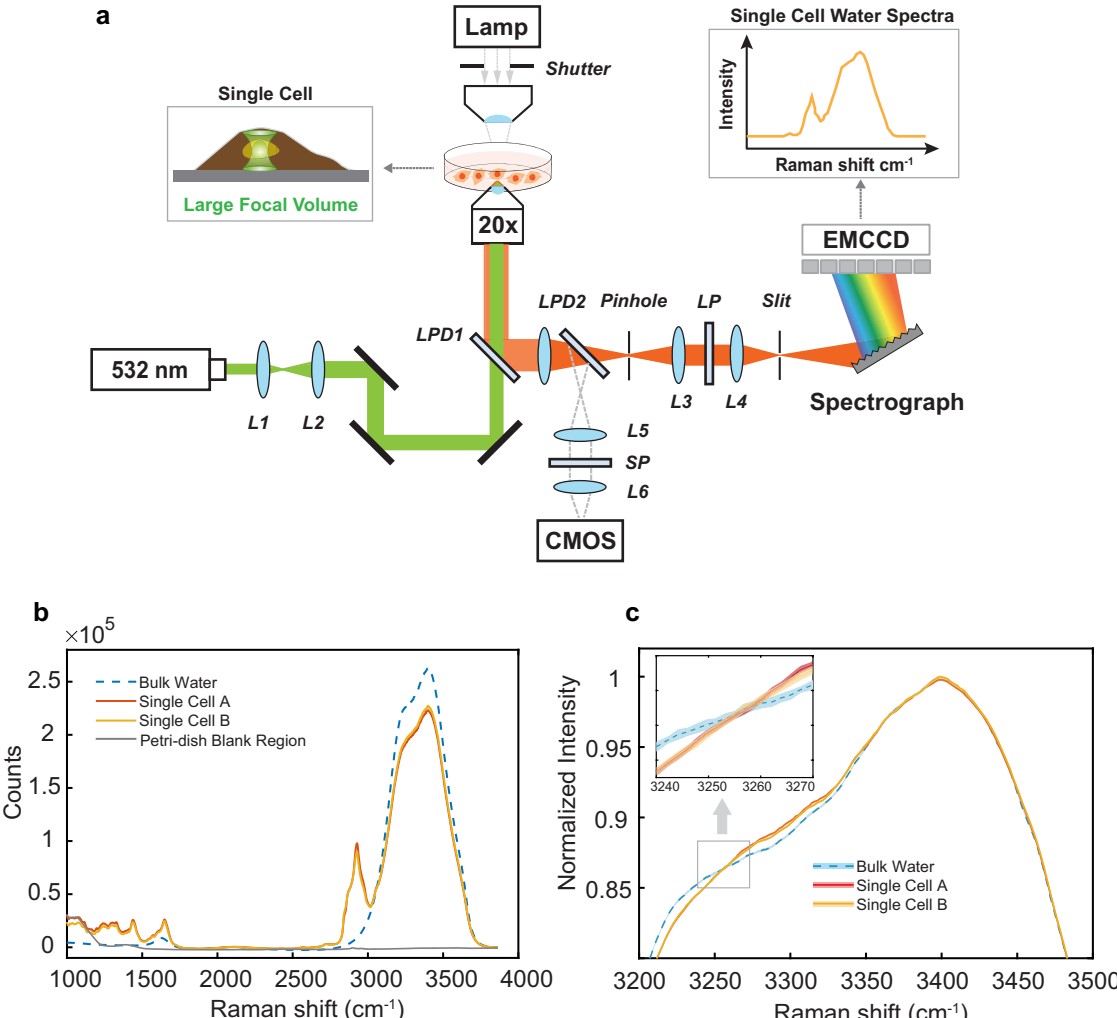

**Fig. 1 | Whole-cell confocal Raman micro-spectroscopy of O-H stretch region.**
**a** Optical schematics of a home-built confocal micro-Raman system. We purposely underfilled the back aperture of a 20× objective (beam diameter is 9–10 mm) to spread the illumination volume (~2 μm × 2 μm × 10 μm) inside a single cell. Details in Supplementary Information. L: lens; LPD1, LPD2: long-pass dichroic beamsplitter; SP: short-pass filter; LP: long-pass filter. L1/L2 ($f$ = 40/300 mm), L3/L4 ($f$ = 150/50 mm), L5/L6 ($f$ = 100/100 mm) (**b**) Comparison of Raman spectra of

representative live HeLa cells and bulk liquid water, obtained at room temperature (24 °C) and 0.1 MPa (1 atm pressure) with 35 mW power on sample during 400 sec acquisitions. **c** Zoom-in view of Raman spectra with normalized 3400 cm⁻¹ peak. A small but non-negligible discrepancy in spectra shape can be identified between live cells and bulk water. Shaded area depicts the standard deviation of 80 repeated measurements of 5-sec spectral acquisition.

stretching region. Vibrational spectra of these solutes are not easy to acquire (which will be elaborated further).

Herein we employed a Raman micro-spectroscopy technique to uncover the structural nature of intracellular water. To address the background from solutes, we proposed and validated an experimental strategy to quantify the spectral contribution from interfering solutes of the whole cell. Then, we demonstrated that the O-H stretch region of the live-cell spectrum is not a simple sum of solute and solvent, proving the existence of structurally altered water, which we attribute to biointerfacial water. To distinguish the overwhelming bulk-like water, we harnessed the emerging Raman multivariate curve resolution (MCR) spectroscopy[36,37] and generalized it to living cells. These technical advances together allow us to unveil the structural composition, abundance and vibrational spectra of intracellular water in living cells, for the first time to the best of our knowledge. On one hand, contrary to popular belief, intracellular water turns out to be largely bulk-like and can co-exist with macromolecular crowding; on the other hand, all globular proteins in living cells are found to be working beneath, on average, one full molecular layer of biointerfacial water that is structurally disordered. Furthermore, we observed spectral

invariance of biointerfacial water among different individual cells and distinct cell types, which we postulate might underlie thermodynamic stability of the proteome inside living cells.

## Results

### Spectral similarity and discrepancy between bulk water and live cells

We home-built a whole-cell confocal Raman microscope to examine live cells (Fig. 1a). The setup is featured by intentionally under-filling the back aperture of a 20× objective lens (effective numerical aperture N.A. ~0.4) and still capturing the scattered Raman signal with a large collection efficiency and a confocal pinhole of 40 μm. This configuration yields an optimal confocal volume (~ 2 μm × 2 μm × 10 μm), which is large enough to probe a substantial fraction of the whole mammalian cell, reducing subcellular heterogeneity and potential phototoxicity, and is also small enough to exclude the contribution from extracellular water. Figure 1b shows Raman spectra, $I_{live\ cell\ OH\ region}$, of two live HeLa cells together with $I_{bulk\ water}$ from pure bulk water, obtained at the same experimental condition. The flat baseline is free from auto-fluorescence and substrate scattering, thanks to the tight confocality.

The time series (Fig. S1) suggests that no photodamage or photo-driven processes occurred during the measurement. While the 2800 - 3000 cm$^{-1}$ region contains spectral information from C-H of biomolecules, the broad high-wavenumber region from 3100 to 3800 cm$^{-1}$, referring to the O-H stretching region, is mostly from water. The relative intensities indicate that the intracellular water concentration is slightly lower than that of pure water, consistent with the fact that living cells contain a large amount of water.

Is the spectral shape of $I_{live\ cell\ OH\ region}$ identical to $I_{bulk\ water}$? Fig. 1c presents the normalized spectra. While the overall shapes are similar, a small but non-negligible discrepancy is reproducible in different cells and is clearly above the measurement noise (shaded error in Fig. 1c). Biomolecules inside cells shall contribute to the difference. Notably, N-H stretching from protein backbone and side chains and even DNA/RNA and O-H stretching from carbohydrates are expected to display Raman scattering in 3100–3500 cm$^{-1}$. This can be assigned to the "intracellular solute background". Such solute background can be intrinsically complicated in living cells, making it challenging to unveil the true spectral feature of intracellular water.

## Quantifying intracellular solute background

It has been estimated in several model organisms that the intracellular concentration of N-H groups (mainly from proteins) dominates over the non-water O-H groups by about one order of magnitude[18]. This insight, together with the fact that proteins constitute ~75% of dry weight of Hela cells[38], suggests N-H vibrations of protein as the major contributor to the solute effect (Supplementary Discussion 1). However, it has been challenging to measure the protein N-H spectrum in situ under strictly physiological conditions. Similar to the solute interference, the O-H stretching from water will present the solvent interference for N-H stretching. To avoid water interference, chiral N−H stretch was recorded by chiral SFG from protein at interfaces[39], which is unfortunately difficult to implement in living cells. Presumably, D$_2$O exchange can be used to replace H$_2$O inside cells. However, O-H groups are generated as soon as hydrogen-deuterium exchange occurs (over minutes for protein backbone[40]) between D$_2$O and intracellular active hydrogen (such as N-H groups), again leading to interference with the target N-H group.

We hence propose an alternative method to quantify the solute contribution. We dehydrated intact HeLa cells under vacuum in an isothermal manner, and then acquired the whole-cell Raman spectrum, which should include the collective sum of all interfering solutes. Figure 2a and Fig. S2 shows cellular spectra, $I_{vacuum\ dehydrated\ cell}$, averaged over 20 cells. The consistency between spectra with vacuum dehydration periods of 2 days and 7 days proves that water is thoroughly removed. A prominent and broad peak is detected around 3300 cm$^{-1}$ (also called amide A in Fig. 2b), similar to the reported chiral SFG spectra of N-H from protein at interfaces[39]. Quantitatively, the ratio between this peak and the C-H$_3$ protein peak in $I_{vacuum\ dehydrated\ cell}$ matches with its counterpart in the spectrum of dry bovine serum albumin (BSA), a popular model protein (Fig. S3 and Supplementary Discussion 1). This validates the expectation that the 3300 cm$^{-1}$ peak of cellular solutes largely originates from the N-H group in intracellular proteins. Other factors, such as carbohydrates, DNA/RNA or residual water should be insignificant.

Caution needs to be taken, as changes during cell dehydration might shift the amide A spectrum. We employ a concept of proximity sensor, namely the carbonyl group (C=O) in the amide unit, to quantify such potential shift. There are several considerations. First, the carbonyl group has local proximity to the N-H group in the protein backbone (Fig. 2b), physically away by only one bond. Second, its bond axis is in parallel with the N-H bond. Hence approximately identical electric fields from environment are projected on these two bonds as predicted by Onsager's reaction field theory[41]. Third, the vibrational signature of C=O groups can be readily characterized as the amide I

band without water interference. Thus, the response of amide I can be utilized as a proximity sensor to infer the nearby change experienced by N-H. As shown by Fig. 2c, the amide I band in $I_{vacuum\ dehydrated\ cell}$ is largely conserved compared to live cells with only a blue-shift of 1.4 cm$^{-1}$. Additionally, the amide III band, derived from coupled C-N stretching and N-H bending, shows nearly identical spectra after dehydration. These observations strongly suggest that corresponding spectral change of amide A should also be relatively insignificant after vacuum dehydration. More quantitative conclusion can be reached by studying vibrational solvatochromism[42] of model compounds (N-methylacetamide, butylamine) (Fig. 2d, e, Fig. S4, Supplementary Discussion 2): less than 4 cm$^{-1}$ blue-shift is expected for amide A after vacuum dehydration, much smaller than the peak width. Such insensitivity can be reconciled by the fact that the isothermal vacuum dehydration employed here only causes a gentle perturbation to intracellular protein conformation compared to the harsh denaturation process (Fig. S5). This rationalization also aligns with our own observation on BSA in vitro (Fig. 2f, Fig. S5) and minor conformational changes reported on globular proteins upon dehydration[43,44]. Together, we conclude that $I_{vacuum\ dehydrated\ cell}$ can largely represent intracellular solute contribution of live cells.

## Live-cell water spectrum is not a simple sum of intracellular solute and solvent

With intracellular solute contribution obtained above, we can revisit the spectral discrepancy identified in Fig. 1c. If the solute contribution were the only cause, one would expect to computationally reconstitute the measured $I_{live\ cell\ OH\ region}$ by adding a certain fraction of $I_{bulk\ water}$ to $I_{vacuum\ dehydrated\ cell}$ (Supplementary Discussion 3). However, Fig. 3a, b demonstrates that the reconstitution always fails to reach perfect agreement with the measurement regardless of the added water concentration. For the closest case with 88.0% water (Fig. 3c), the addition of solute and solvent is higher than live-cell spectrum in the red end and lower in the blue end, respectively. Even in the hypothetical scenario with solute spectra that are spectrally shifted or amplitude modulated, the reconstitution still cannot be achieved (Fig. S6). This study proves that the live-cell spectrum is not a simple sum of the solute and the solvent. Thus, there must exist a third component that is spectrally altered from that of bulk water.

## Raman MCR micro-spectroscopy uncovers biointerfacial water inside living cells

Numerous studies have shown that water structure and dynamics are altered in the close vicinity of biomolecules. We thus attribute the spectrally distinct third component inside living cells to biointerfacial water. How to distinguish biointerfacial population from bulk-like water is another challenge. To achieve the in-solution interface selectivity, we adopt the emerging concept of Raman-MCR spectroscopy[36,37,45–47]. Raman-MCR does not have any assumption about the amplitude, position, and width of spectral bands (except the absence of negative concentration). By computing the non-negative minimum-area difference between the solution and pure solvent spectra, it can separate the contribution of bulk water and unravel the properties of water in the hydration shell of small solutes. Raman-MCR has been primarily used in small molecules such as alcohols and ions with relatively large hydration shells and hence negligible solute contribution[36]. In parallel, a related approach has been utilized whereby MCR is combined with FITR spectroscopy to analyze solvation shell of antifreeze proteins in vitro[48]. Here, we generalize it to live-cell Raman spectrum (Supplementary Discussion 5), which has to include the additional intracellular solute background (Eq. 1):

$$I_{live\ cell\ OH\ region} = I_{vacuum\ dehydrated\ cell} + I_{biointerfacial\ water} + x\,I_{bulk\ water}$$

(1)

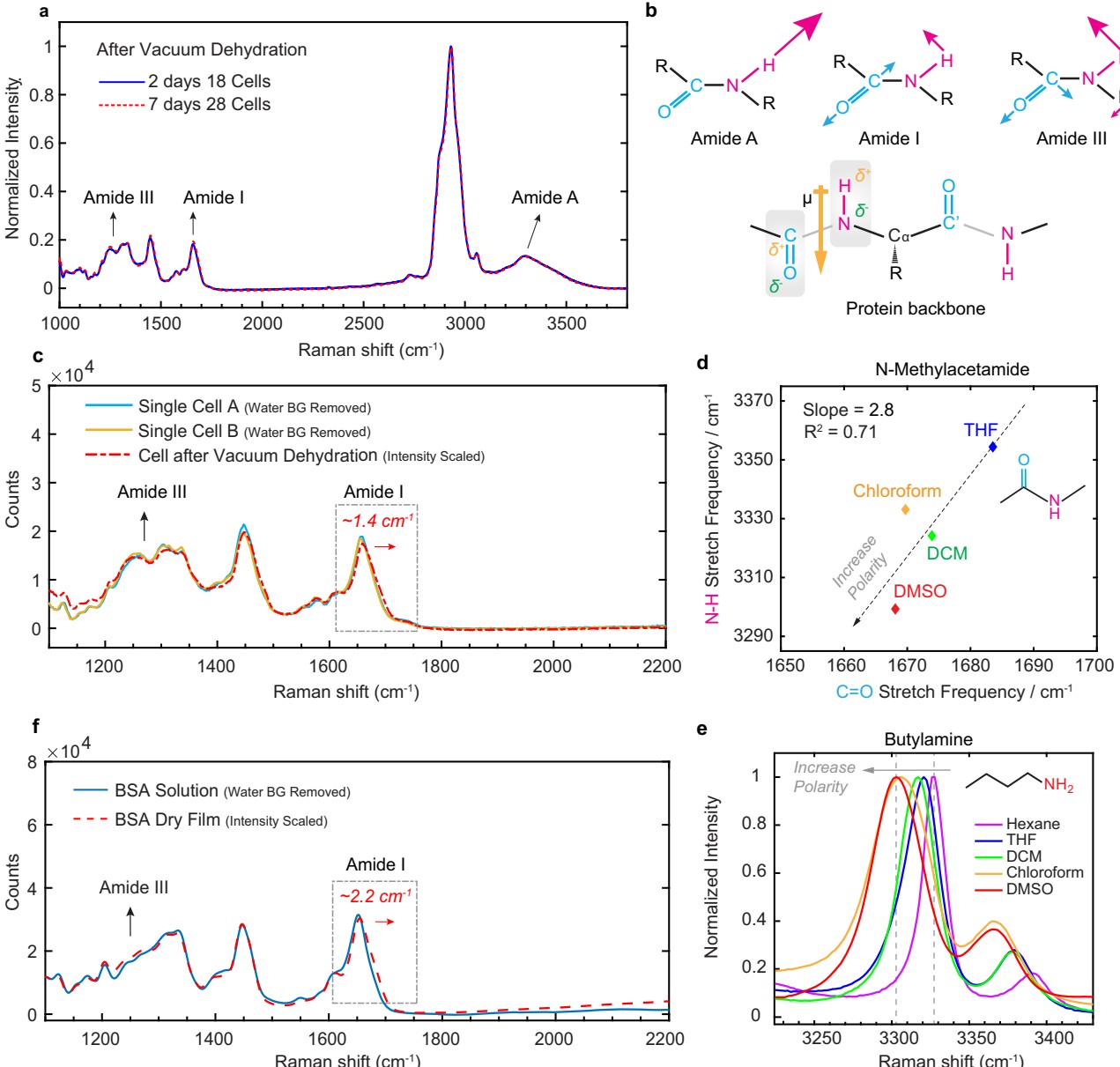

**Fig. 2 | Quantification of intracellular solute background in O-H stretch region.**
**a** Averaged Raman spectra of HeLa cells after vacuum dehydration, normalized to the CH₃ peak (2930 cm⁻¹). Blue: 18 cells measured after 2-day dehydration. Red: 28 cells measured after 7-day dehydration. **b** Top: schematics of the main vibrational modes of peptide bonds in Raman spectroscopy. Bottom: Illustration of the "proximity probe" concept in protein backbone. **c** Comparison of Raman spectra of live HeLa cells (blue and yellow line) and 7-day vacuum dehydrated HeLa cells (red dash line), showing a blue-shift about 1.4 cm⁻¹ of amide I peak after dehydration. **d** Vibrational solvatochromism studies of C=O and N-H groups in

N-methylacetamide (NMA), a model compound for amide. NMA is measured at 1% (v/v) in different solvents. The slope of linear relationship, which corresponds to the ratio of the Stark tuning rates: $|\triangle\vec{\mu}_{N-H}|/|\triangle\vec{\mu}_{C=O}|$, is estimated to be 2.8. **e** Vibrational solvatochromism of butylamine, a model compound for N-H group in protein side chains, showing frequency shift and peak broadening in more polar solvents. **f** Comparison of Raman spectra of BSA solution (blue line) and BSA dry film (red dash line), showing a blue-shift about 2.2 cm⁻¹ of amide I peak after dehydration.

High quality spectrum, $I_{bulk\ water}$, was acquired with an over 1000:1 signal-to-noise ratio (Fig. S7). Figure 3d maps the MCR residual analysis by subtracting $I_{vacuum\ dehydrated\ cell}$ (after scaling based on the dominant CH peak at 2930 cm⁻¹) and a varying fraction (x%) of $I_{bulk\ water}$, from $I_{live\ cell\ OH\ region}$. A critical point is marked at $x = 83\%$ (horizontal dash line); at this point, the difference spectrum from 3054 cm⁻¹ to 3150 cm⁻¹ simultaneously reaches the vanishing threshold, characterized by a continuously flat green line (Fig. 3d). Crossing over this point, the difference spectrum starts to display non-physical negative values (Fig. S8). This collective behavior not only guarantees the non-negative minimum-area condition to be

fulfilled in Raman MCR algorithm but also serves as an internal check.

Figure 3f depicts $I_{live\ cell\ OH\ region}$, $0.83 \times I_{bulk\ water}$ and their difference which is named solute-correlated (SC) spectrum. Note that the SC spectrum nearly overlaps with $I_{vacuum\ dehydrated\ cell}$ from 3054 cm⁻¹ to 3150 cm⁻¹, dictated by the critical point $x = (83\%)$ identified by Raman-MCR. Finally, according to Eq. (1), the difference between the SC spectrum and $I_{vacuum\ dehydrated\ cell}$ retrieves $I_{biointerfacial\ water}$ (Fig. 3f). While the retrieval of $I_{biointerfacial\ water}$ is sensitive to the quality of $I_{bulk\ water}$, it only changes slightly upon spectral shifts of $I_{vacuum\ dehydrated\ cell}$ (Fig. S9). Similar analyses on two other individual

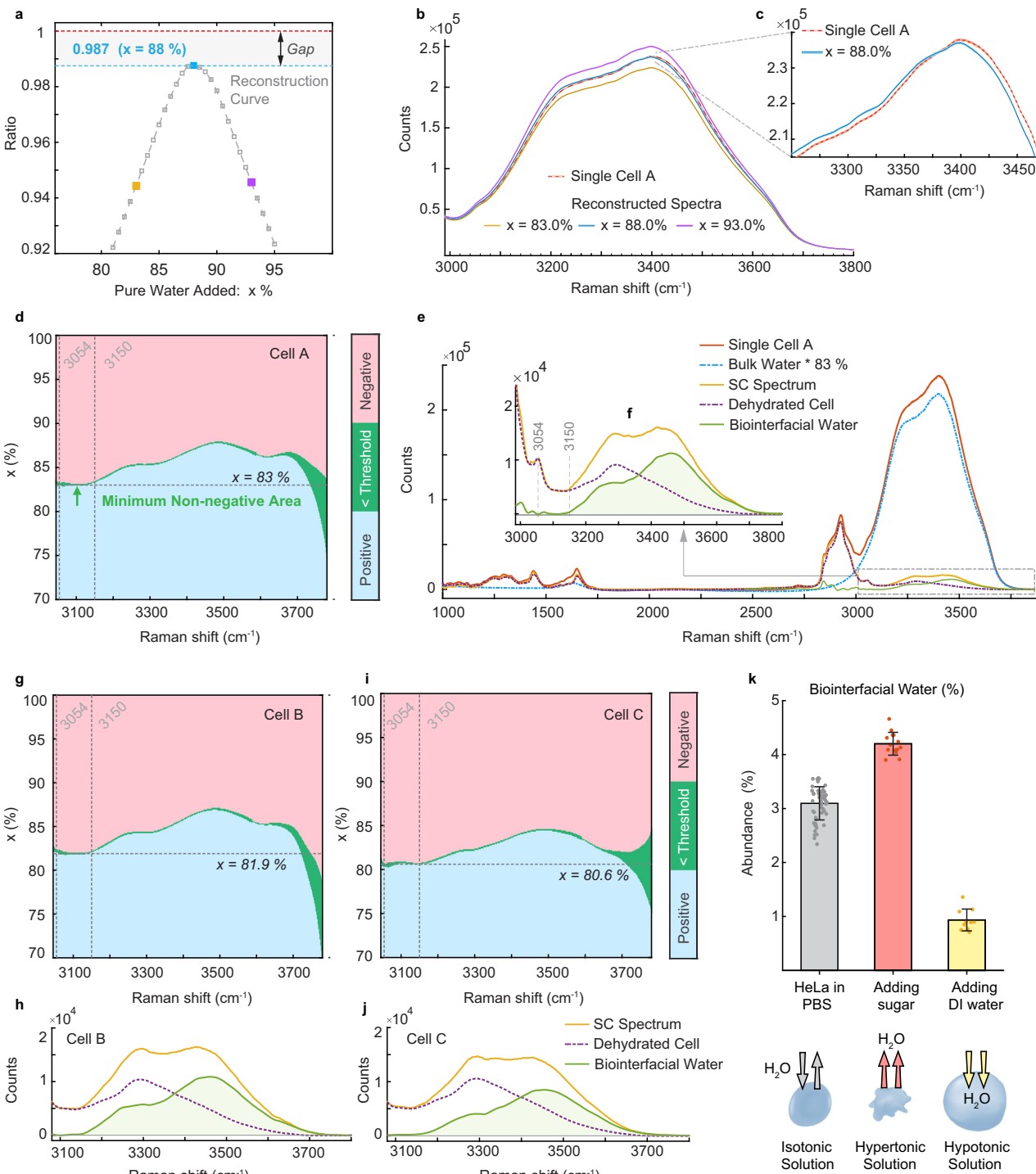

**Fig. 3 | Identification of biointerfacial water in live cells using Raman-MCR spectroscopy. a** Mathematical reconstruction of live-cell water spectra by adding a varying fraction of bulk water spectrum and a scaled dehydrated cell spectrum, following Eq. (S2). The ratio between the integrated area and the experimental result never reaches 1. **b** Measured single-cell spectrum and reconstructed cell spectra with different bulk water contributions (x%). **c** Zoom-in view when the addition is optimized at x = 88%, showing that the two-component reconstruction cannot reproduce the exact spectral profile of O-H stretch region of living cells. **d** Residual map of Raman-MCR analysis after subtracting varying amounts of bulk water (x%) and dehydrated cell spectrum from live-cell O-H spectrum. The transition boundary is highlighted as light green when the absolute value of the residual is less than a threshold (defined as 0.1% of single-cell spectrum intensity here). The critical point is defined when the minimum non-negative area is achieved, in this

case, when x = 83%. For bulk water contribution >83%, the differential SC spectrum turns negative in the O-H stretch region (non-physical event), and for bulk water contribution <83%, the SC spectrum does not represent the spectrum with minimum area. Details in Supplementary Discussion 5 and Fig. S8. **e** Raman spectra after subtracting 83% of bulk water and the scaled dehydrated cell spectrum from the live-cell spectrum (red), identifying the residual as biointerfacial water (green), based on Eq. (1). Solute-correlated spectrum (yellow), obtained when only removing the bulk water contribution (blue dash) from live cell (red). **f** zoom-in view of O-H region. **g–j** Further analysis and spectra from two other HeLa cells. **k** Analysis of biointerfacial water abundance (% in total intracellular water) at different osmotic conditions [PBS (n = 57), 300 mM mannitol solution (n = 14), deionized water (n = 10)], shown as mean ± std. Details in Fig. S10.

HeLa cells are shown in Fig. 3g–j, exhibiting a consistent spectral shape.

Raman scattering cross section of liquid water is generally considered relatively insensitive to changes in hydrogen-bonding strength and structure[46,47,49]. By comparing the integrated spectral area with that of pure bulk water (55.5 molar), we conclude that biointerfacial water and bulk-like water are 1.4 molar and 46 molar, respectively, and that biointerfacial water constitutes ~3% of the total intracellular water (~47.4 molar). We further show that osmosis perturbations (such as adding pure water or sugar to the cell culture media) can evidently regulate the composition of intracellular water (Fig. 3k, Fig. S10), proving the physiological relevance of our measurement. To the best of our knowledge, it is the first time that the structural composition, abundance and vibrational spectra of intracellular water are experimentally determined.

### Structural features of biointerfacial water: weakened hydrogen-bonding, more disordered structure and presence of dangling O-H group

We next analyze the retrieved $I_{biointerfacial\ water}$ to learn the underlying structural features. To make a more informed comparison, we measured and reproduced the hydration shell spectra around the hydrophobic groups of two small molecules [*tert*-butyl alcohol (TBA) and ethanol] (Fig. 4a)[37,50]. In general, strong hydrogen-bonding between adjacent water molecules weakens the covalent O-H bond, causing O-H to shift towards lower vibrational frequency (i.e., red shift). We hence quantify the frequency shift by calculating the average O-H frequency as $\langle\omega\rangle = \int_{\omega_1}^{\omega_2} \omega I(\omega)d\omega$ where $I(\omega)$ is the corresponding intensity when the band shape is normalized to the unit area between $\omega_1 = 3100$ cm$^{-1}$ to $\omega_2 = 3800$ cm$^{-1}$. Thus, a blue-shift (~36 cm$^{-1}$) of O-H from biointerfacial water compared to bulk water suggests a weakened hydrogen-bonding network, which is opposite to the red shift of hydration shell around two small alcohols (Fig. 4b).

Errington-Debenedetti tetrahedral order parameter (TOP) is a popular scalar used to quantify the structure of water molecule, with 0 and 1 representing ideal gas and perfect tetrahedron, respectively[51]. TOP has been attributed to cause the spectral shift of $\langle\omega\rangle$ of water[52,53], and a nearly linear correlation between these two quantities was discovered with a slope of −0.00149 (Fig. S11)[54–56]. Given that TOP of pure water is about 0.66 at 24 °C and a blue-shift of 36 cm$^{-1}$ observed here, TOP = 0.57 is calculated for biointerfacial water (Fig. 4b), a nearly 14% reduction. It is interesting to note that the average TOP of the hydration shell of a few proteins have recently been calculated by molecular dynamics simulations, yielding 4-8% reduction compared to bulk water[57–59].

The broad O-H spectrum originates from different populations of water structure. Particularly, two significant populations are categorized as strongly hydrogen-bonded (i.e., ice-like) and weakly hydrogen-bonded (i.e., liquid-like) water, respectively[53,60–62]. The peak ~3250 cm$^{-1}$ is typically assigned to the former and the peak ~3450 cm$^{-1}$ is often attributed to the latter. $I_{biointerfacial\ water}$ can be readily fit by a sum of these two peaks (Fig. 4c), supporting its water origin. Compared to the bulk water or hydration water around alcohols whose $A_{3250}/A_{3450} > 1$, the area ratio of $A_{3270}/A_{3460}$ ~ 0.17 is markedly smaller in biointerfacial water, characterizing a more liquid-like structure. Moreover, these two populations have been associated with TOP = 0.78 (mostly with 4 hydrogen bonds) and TOP = 0.52 (mostly with 3 hydrogen bonds), respectively[52,53,63–65]. Applying the relative weights of 0.17 and 1 to these two populations, an averaged TOP can be calculated as 0.56 (very close to the analysis above) for biointerfacial water which shall have an average of 3.1 hydrogen bonds.

One may take closer look when interpreting the molecular basis of the spectral change of O-H stretch[66,67]. It is noted that the observed suppression of the 3250 cm$^{-1}$ band relative to the 3450 cm$^{-1}$ band might not only signify a transformation towards a weaker hydrogen-

bonding network in the vicinity of biomolecules but could also reflect reduction in the collective vibrational coupling within the water matrix. Such change lessens the delocalization of vibrational energy across multiple water molecules, potentially leading to decreased spectral response at 3250 cm$^{-1}$ and a blue-shift in the central frequency of the O-H stretch, compared to bulk water. Isotopic dilution techniques, which have been effective in differentiating between the impacts of hydrogen-bonding perturbation and vibration coupling in studies of ion-induced and hydrophobic hydration shells, face considerable challenges in cellular context[68,69]. Despite the complexity of molecular basis of spectral change, both the weakening of hydrogen bonds and the reduction in vibrational coupling converge on a consistent structural theme: a more disordered structure in biointerfacial water.

Interestingly, we also observed a small peak around 3650 cm$^{-1}$ in some cells (Fig. 4d). This high-frequency peak is often attributed to non-hydrogen-bonded free (dangling) O-H group, a common spectral feature of hydrophobic hydration[47,70,71]. While the peak position here is similar to that of dangling O-H in the hydration shell around small hydrophobic alcohols, its peak width (~90 cm$^{-1}$) is significantly broader (Fig. S12), suggesting higher heterogeneity of intracellular environment.

### Whole-cell model of biointerfacial water: one water layer hydrating cellular proteins

It is constructive to analyze the molecular origin of the detected biointerfacial water. Many species of small electrolytes exist inside a live cell (Table S1). It has been observed that monovalent cations such as Na$^+$ and K$^+$ have a negligible effect on the O-H stretching band of hydration shell[68,72]. Regarding anions, phosphate is the most dominating species. However, most phosphates are associated with macromolecules as organic forms. The rest of the phosphate is in free form (indicated as Pi), estimated at 40 mM. Given the total amount of anions (60 mM, including Pi, Cl$^-$ and HCO$_3^-$), the affected water population is estimated to be 240 mM, assuming there are four water molecules in the hydration shell per anion. Hence, mobile electrolytes can only account for a small fraction of the biointerfacial water (~1.4 M). The main contributor shall be macromolecules. Considering that proteins constitute about 75% of dry weight of Hela cells and DNA/RNA only accounts for ~10%[38], we are prompted to build a model where biointerfacial water is mostly located in the hydration shell of proteins (Fig. 4e).

More quantitatively, we can estimate the thickness, $D$, of the biointerfacial water in a single-cell as

$$D_{biointerfacial\ water} = \frac{N_{interfacial\ water\ per\ protein} \cdot V_{H_2O}}{\text{Solvent accessible surface area}} \qquad (2)$$

in which $N$ is the average number of interfacial water molecules associated with one globular protein, calculated as the concentration ratio (~1500) between biointerfacial water (1.4 M) and globular protein (0.94 mM), $V_{H_2O}$ is the volume of one water molecule (~30 Å$^3$), and the solvent-accessible surface area (SASA) is approximately 18,000 Å$^2$ for an average human protein. Details of these estimations can be found in Supplementary Discussion 4 and Fig. S13, 14.

Equation (2) yields a thickness value of 2.6 Å, corresponding to just one molecular layer of water. Importantly, a view of short-range protein hydration is emerging in the literature[58,73–76]. In particular, a systematic computational study of the spatial range of protein hydration reveals all investigated structural and dynamical properties have exponential decay lengths of less than one hydration shell[58]. Since rotational ambiguity renders Raman-MCR algorithm to give the lower bound for the abundance of biointerfacial water[46], the combination with the upper bound of just the first hydration shell (strongly suggested by the computational study[58]) encourages us to quantify the

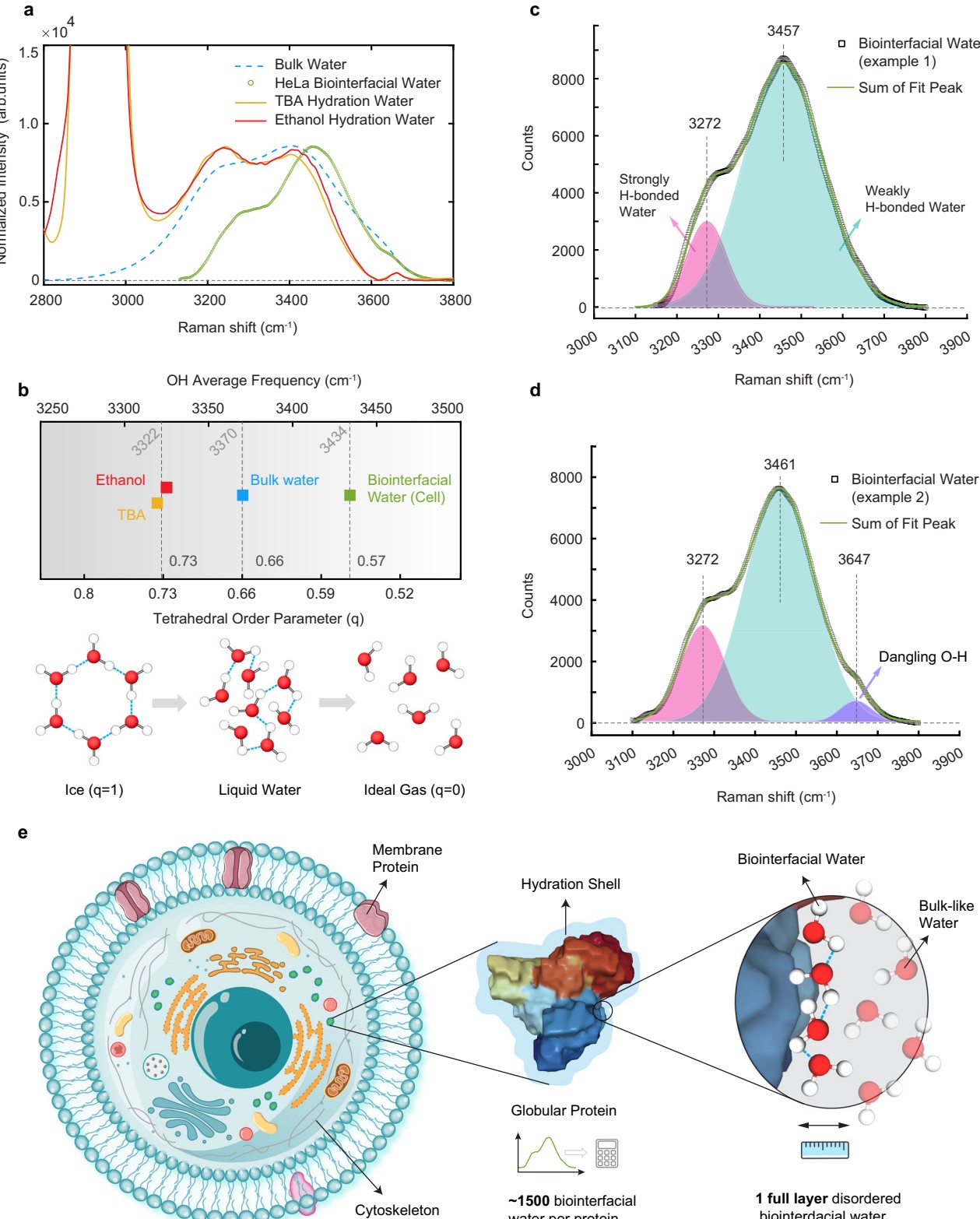

**Fig. 4 | Structural features of biointerfacial water. a** Comparison of Raman spectra of bulk water (blue dash), intracellular biointerfacial water (green) and hydration water of alcohols [tert-butanol, (TBA, yellow) and ethanol (red line)]. **b** Averaged O-H frequency and the corresponding tetrahedral order parameter of four species in **a**. The tetrahedrality (q) of water decreases from 0.66 for bulk water at 24 °C to 0.57 for biointerfacial water, estimated from a frequency difference of 36 cm⁻¹. Details in Fig. S11. **c**, **d** Biointerfacial water spectra from two representative

HeLa cells. Spectra were fit to a sum of Gaussian peaks to extract the areas under the distinct O-H peaks including strongly hydrogen-bonded (pink area) and weakly hydrogen-bonded (cyan area) populations of water. Dangling O-H peak can be observed in single-cell spectrum shown in **d** (purple area). **e** Whole-cell model of biointerfacial water derived from our experimental results: one full layer of structurally altered water molecules surrounding the water-solvable globular proteins inside a living cell.

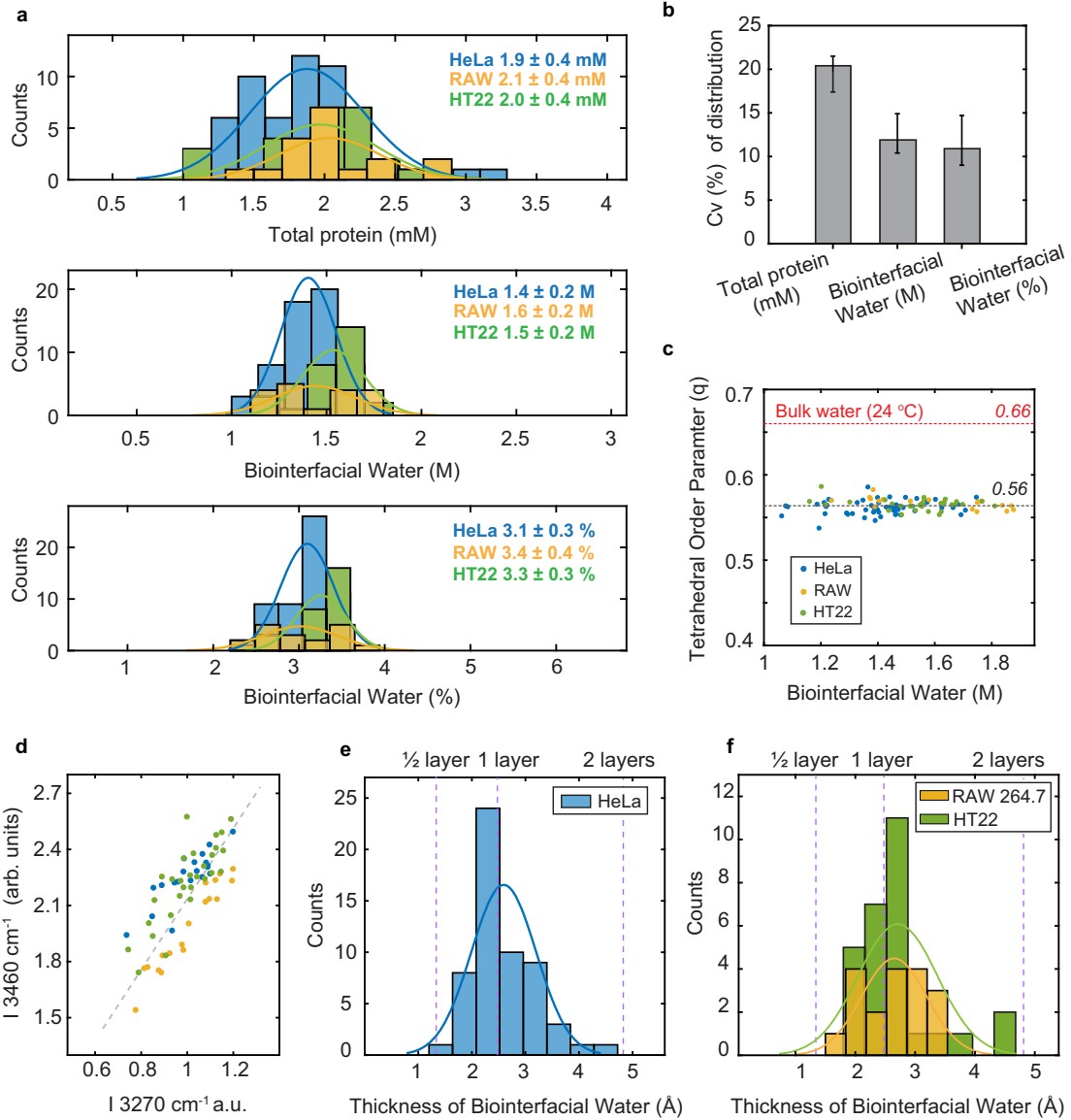

**Fig. 5 | Single-cell water spectroscopy. a** Distributions of total protein concentration, amount of biointerfacial water (molar), weight of biointerfacial water (%, as compared to intracellular water) in different cell lines. **b** Heterogeneity of three parameters presented in **a**. Coefficient of variation ($C_V$) values for each parameter were calculated as the ratio of the standard deviation to the mean for all cells including HeLa (57 cells), Macrophage RAW 264.7 (18 cells) and HT22 (28 cells). Total sample size $n = 103$. Error bars represent the minimum and maximum of values. **c** Plot of tetrahedral order parameter as a function of biointerfacial water concentrations, showing the invariance of water tetrahedrality (mean ± s.d. is 0.56 ± 0.01). **d** Positive correlation between the intensities of the two main peaks in biointerfacial water spectra. **e, f** Calculated thickness of biointerfacial water in different cell lines based on the whole-cell model. HeLa: $D = 2.6 \pm 0.6$ (Å). RAW 264.7: D = 2.7 ± 0.7 (Å). HT22: D = 2.7 ± 0.5 (Å).

biointerfacial water as one full water layer hydrating intracellular proteins (Fig. 4e).

### Single-cell water spectroscopy reveals invariance of biointerfacial water

To gain general insights, single-cell studies are conducted among 57 HeLa cells and two other cell lines (mice RAW 264.7 macrophages and HT22 neuronal cells, 46 cells in total). While the measured protein concentration shows a wide range of distribution, biointerfacial water is surprisingly conserved. First, its abundance levels are very similar across three cell types and each display a tight distribution (Fig. 5a, b): 3.1 ± 0.3% (HeLa cells), 3.4 ± 0.4% (RAW 264.7) and 3.3 ± 0.3% (HT22). Second, the structural parameter TOP, as deduced from the average frequency of O-H, is bound to a mean value ~0.56 for all three cell types (Fig. 5c). Third, spectral shape of $I_{biointerfacial\,water}$ bears similarity

among different cells, as indicated by the strong linear correlation (Fig. 5d) between the 3270 cm$^{-1}$ peak and the 3460 cm$^{-1}$ peak. Hence, a weakened hydrogen-bonded network and a lower TOP seem to be general features for biointerfacial water. Fourth, the thickness of biointerfacial water also exhibit a relatively narrow distribution among individual cells and cell types. For HeLa cells, the thickness spreads from half layer to nearly two monolayers of water, with an average value of nearly one molecular layer (Fig. 5e). Similar distributions were observed in RAW 264.7 and HT22 cells (Fig. 5f). Hence, we uncover the generality and invariance of biointerfacial water in living cells.

## Discussion

Intracellular water structure had remained elusive. While vibrational spectroscopy can interrogate the hydrogen-bonding network of water, living cells present a formidable challenge for achieving the necessary

selectivity. Specifically, how to avoid the complication of interfering solutes and how to distinguish biointerfacial water from abundant bulk-like water are two hurdles. The methods we employed here, particularly quantifying cellular solutes after isothermal vacuum dehydration and live-cell Raman-MCR spectroscopy, overcome these challenges and thus open up experimental possibilities of interrogating water structure in vivo. Our present study attempts to obtain a spatiotemporally *averaged* picture of intracellular water with a high signal-to-noise ratio, as this will form the foundation of the missing knowledge. This is why we built a whole-cell Raman micro-spectroscopy instrument to integrate over time (~ 400 sec) and space (probe volume > 10 $\mu m^3$, as opposed to ~0.1 $\mu m^3$ of standard confocal Raman microscope) inside individual live cells. However, we expect that our methodology can be extended to other vibrational modalities with varying length and time scales[77–79]. Such explorations promise to unveil water structure in vivo with high spatial (e.g., resolving various organelles) and temporal resolution.

Is intracellular water bulk-like or not? Two schools of thought have persisted in the literature. The first perspective takes the notion of macromolecular crowding (up to 40% of the cell's volume is occupied by macromolecules[80]) into consideration, and suggests that most water in cytoplasm behaves very differently from bulk water and little to no "bulk-like" water is present within cellular environments[19–21]. In fact, Pollack et al. state in the preface that "practically all cell water is interfacial"[22]. Recent reviews further argued that, the average distance between macromolecules in the cytoplasm is around 1 nm, corresponding to just three to four molecular layers of water—which, based on classical solvation theory, cannot be considered bulk-like[23–25]. On the other hand, a series of experiments reported that water dynamics in cells is largely similar to that of pure water with a small fraction exhibiting slower dynamics[8,10,14,18]. Our result of intracellular water composition clearly supports that intracellular water is largely bulk-like. Note that our conclusion is based on structural features probed by vibrational spectroscopy, while the previous experiments relied on dynamical observables. As different properties of water respond differently to the presence of an interface[81], this might account for the different bulk-like percentage between our result (~97%) and the literature (80–90%)[8,10,14,18]. Different types of cells under investigation could be another factor.

Then how to reconcile with the notion of macromolecular crowding? First, macromolecular crowding is mainly an effect exerted by large molecules on the properties of other large molecules. Small molecules such as small solutes and solvents are less affected. Second, a substantial fraction of the cell's volume occupying by macromolecules does not necessarily lead to a close distance between macromolecules, as the sizes of macromolecules themselves are generally large. Third, the actual distance between macromolecules in the cytoplasm might not be as small as believed (~1 nm). For HeLa cells, our Raman measurement indicates an average protein concentration of 1.87 mM (Supplementary Discussion 4), very close to the reported 1.6 mM by proteomics[82]. The average separation of proteins can be calculated to be ~10 nm. Given the average 4-nm protein size, the edge-to-edge distance between proteins is thus ~6 nm. Hence, macromolecular crowding and largely bulk-like intracellular water can co-exist.

Although biointerfacial water only occupies ~3% of the total intracellular water, it would be mistaken to neglect its importance. It can reach 1.4 M, making it much more concentrated than the most abundant electrolyte in the cell ($K^+$ is only 0.1 M). Besides its high concentration, this population of water resides at biointerface to interact with macromolecules, mediating or even governing many vital biological processes. Its crucial location is captured by our model (Fig. 4e): averagely speaking, all the globular proteins inside live cells are functioning underneath a full molecular layer of biointerfacial water. Its thickness is an interesting matter. In fact, the spatial range of

the protein-induced perturbation of water structure and dynamics has been a contentious issue. Several studies support long-ranged perturbation in protein hydration shell[16,83,84]. In contrast, a NMR experiment has indicated that the perturbation is short-ranged and mostly limited to the first hydration shell[73]; a neutron scattering experiment found less than two shells of water are dynamically affected by green fluorescent protein in solution[74]; a recent chiral SFG study finds structural perturbation persists only into the first hydration shell of proteins[76]; a comprehensive simulation concluded that the solvent perturbation is short-ranged as a consequence of the high energy density of bulk water, with all investigated structural (including TOP) and dynamical properties having exponential decay lengths of less than one hydration shell[58]. Therefore, our result of biointerfacial water surrounding the globular protein with less than two molecular layers (Fig. 5e, f) aligns well the short-range protein hydration picture. It is remarkable for in vivo result to be corroborated by in vitro measurement and computational results.

Vibrational spectrum of biointerfacial water in living cells is a key finding of our study. It would be constructive to compare this in vivo result with the protein hydration shell in vitro. Unfortunately, although SFG spectroscopy has been applied to study interfacial proteins[85], vibrational spectra of protein hydration shells are relatively scarce, and concerns have been raised regarding the impact of air/water interface on structure and charge profile of protein[86]. Nevertheless, molecular dynamics simulations have provided useful references, albeit without full spectra. The mean TOP of the hydration shell of an antifreeze protein is ~4% lower than that of bulk water[57]. TOP around a globular protein, 1A4V, is lower than that of bulk water, by about 8%[59]. TOP averaged over the first shell of small globular proteins is 4.4% smaller than in bulk water with positive (8.5%) and zwitterionic (13%) residues giving rise to the largest reduction[58]. Hence, our result of lower TOP (about 14% compared to bulk water) of the biointerfacial water in living cells is in semi-quantitative agreement with these simulations. Moreover, the average number of hydrogen bonds formed by first hydration shell water molecules has been computed to be 3.1-3.2[76], which agrees well with 3.1 hydrogen bonds estimated for biointerfacial water.

Importantly, vibrational spectra of biointerfacial water are very similar among three distinct cell types as well as different individual cells (Fig. 5 and Fig. S15). Even considering the averaging effect over all proteins inside cells, such spectral invariance might still be surprising, because the hydration structure results from a complex (in a non-additive manner) interplay between nanometer-scale chemical heterogeneity and surface topology of macromolecules[87–89]. In fact, simulations have observed broad distributions and heterogeneous character of protein hydration shell[90,91]. Whether the protein surface topography and chemical heterogeneity have evolved to produce the invariant character of biointerfacial water is an interesting question and remains to be determined.

## Methods

### Home-built whole-cell confocal Raman micro-spectroscopy
The schematics of our home-built whole-cell confocal Raman microscope are shown in Fig. 1a. A 532 nm laser (Samba 532 nm, 400 mW, Cobolt Inc.) is used as the light source. The laser beam was first collimated and expanded by telescope lenses [L1 ($f$ = 40 mm), L2 ($f$ = 300 mm), Thorlabs] to achieve the designed illumination spot size. The expanded beam was then directed to an inverted microscope (IX71, Olympus) installed with a dichroic beamsplitter (LPD1, LPD02-532RU-25, Semrock). A microscope objective (UplanSApo ×20, N.A. = 0.75, Olympus) was underfilled, and the resulting illumination spot had a dimension of ~2 $\mu m$ × 2 $\mu m$ × 10 $\mu m$ inside living cells. The emitted Raman signal first passed a pinhole (PH, 40 $\mu m$, Thorlabs) for background suppression and was relayed by two lenses (L3, L4, Thorlabs) before being projected to the spectrometer (Kymera 328i with 600 lines/mm grating blazed at 500 nm, Andor). A long-pass filter

(LP, LP03-532RU-25, Semrock) was installed between the relay lenses to block laser light from Rayleigh scattering. Raman signal was then collected by an EMCCD (Newton970, Andor). For brightfield imaging, a long-pass dichroic (LPD2, FF511-Di01, Semrock) was installed in front of the pinhole, and a set of relay lenses (L5, L6) were used to project the brightfield images to the CMOS camera (DCC1645C, Thorlabs). A short-pass filter (SP) was installed to suppress ghost images. All the data collection was performed with a custom LABVIEW program (National Instruments). Prior to the preprocessing procedures, the spectra were calibrated by 1:1 mixture of acetonitrile and toluene taken on the same day.

## Estimation of illumination volume

The diffraction-limited spot sizes, both laterally and axially, are calculated using a simplified equation derived from the more complex Airy disk formula. In our experimental setup, with a wavelength ($\lambda$) of 532 nm, and an effective numerical aperture (N.A.) ranging from 0.3 to 0.4, determined by the beam's dimensions at the objective lens's back aperture (20×, air), we estimated the confocal volume to be ~2 μm × 2 μm × 10 μm.

$$d_{xy} = \frac{1.22 * \lambda}{N.A.} = [1.6 \sim 2.2]\,\mu m$$

$$Z = \frac{2 * n * \lambda}{(N.A.)^2} = [6.65 \sim 11.8]\,\mu m$$

## Baseline correction in spectra data

The baseline correction was performed by subtracting the background intensity at 3600 cm$^{-1}$, without applying further corrections or normalization.

## Spectral analysis

Details of Raman-MCR analysis can be found in Supplementary Discussion 5 and Fig. S8.

## Cell culture

HeLa, RAW 264.7 and HT22 cells were cultured in Dulbecco's Modified Eagle's medium (DMEM: 4.5 g/L glucose; 11965, Gibco) supplemented with 10% Fetal bovine serum (FBS; 16000, Gibco) and 1% antibiotics (Penicillin-Streptomycin-Glutamine; 10378016, Gibco). Cells were maintained at 5% $CO_2$ at 37 °C. Cells were first seeded onto the petri dish (14 mm glass bottom precoated with poly-lysine; P50GC1.514 F, MatTek Corporation), and then cultured for 12 h before experiments. Then cells were washed with DPBS (14040, Gibco) and immediately used for Raman spectroscopic measurements at room temperature within an hour. The laser power on sample was tuned in the range of 20-50 mW. Acquisition time ranged from 300 s to 400 s. For vacuum-dehydrated cells, cells (seeded on a petri dish) were first washed with DPBS and put into a vacuum desiccator without any additional fixation for 2–7 days. For osmotic pressure experiments, HeLa cells (seeded on a petri dish) were first washed with DPBS, and the imaging media was replaced with sugar solution (300 mM Mannitol) or deionized (DI) water to control the water flow through the cells. The hypertonic solution causes the cell body to shrink, and the hypotonic solution causes the cell to swell and eventually burst. Under the hypertonic condition, Raman measurements were taken with a 60× objective lens to maintain the confocal volume inside the cells.

## Materials

N-Methylacetamide (Sigma #26305); Butylamine (Sigma #471305); Dimethyl Sulfoxide (VWR #VN182); Tetrahydrofuran (Sigma #401757); Chloroform (VWR BDH1109); Dichloromethane (Sigma #270997); BSA (Sigma #A9418).

## Reporting summary

Further information on research design is available in the Nature Portfolio Reporting Summary linked to this article.

## Data availability

The data that support the findings of this study are available from the corresponding author upon request, and are provided in the Source Data file. Source data are provided with this paper.

## Code availability

The source code for data analysis and example data for testing is publicly accessible at GitHub repository [https://github.com/Sonya922/MinLab_CellWaterRamanMCR].

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

## Acknowledgements

We thank Ethan Perets, Martin Zanni, Naixin Qian and Biman Bagchi for their helpful discussions. We acknowledge support from the Multi-University Research Initiative (MURI) of the Air Force Office of Scientific Research (FA9550-21–1-0170) and the National Institute of Health (R35 GM149256).

## Author contributions

X.L., L.S and W.M. designed the project. X.L. conducted the experiments and analyzed the data. Z.Z. assisted with the experimental setup. X.L. and W.M. wrote the manuscript. All authors contributed to insights through extensive discussion.

## Competing interests

The authors declare no competing interests.
