## [Peer Review File · Nature Communications]

Reviewers' Comments:

Reviewer #1:

Remarks to the Author:

Water intricately regulates biological processes, yet the structure of intracellular water remains enigmatic. This study addresses critical questions about differences in the hydrogen-bonded network within living cells compared to bulk water, exploring its abundance, location, and new structural features. To overcome the experimental challenges, the authors used confocal Raman microscopy and MCR analysis along with the reference Raman spectrum of vacuum dehydrated cell components. The results suggest a consistent ($\sim 3\%$) population of structurally-altered non-bulk-like water near biomolecules. It exposes a weakened hydrogen-bonded network and heightened disorder. Statistical constancy across cells could be an indication of a universal biophysical role of such interfacial water, marking this research as a gateway to in vivo water structure exploration. The paper would be of interest to the readership of the journal, *Nature* and may be accepted for publication and careful consideration of the following concerns.

1. The blue shift of the central frequency of the OH stretch band is taken as an evidence of weaker H-bonding of the perturbed intracellular water. However, the analysis of the OH stretch band of water is not so straightforward due to the presence of intra and intermolecular coupling. In fact, the 3250 cm^{-1} band of the OH stretch spectrum is not solely due to strongly H-bonded water, but mainly due to collective nature of the OH-stretch vibration of water. The collective vibration of water is affected at interface and in the hydration shell of a solute or macromolecule and it may either increase or decrease depending on the nature of the solute (*J. Chem. Phys.* 128, 224511 2008; *Phys. Chem. Chem. Phys.*, 2010, 12, 982–991; *J. Mol. Liq.* 390 (2023) 122987; *J. Phys. Chem. B* 2013, 117, 16479–16485). Therefore, the spectral change of the intercellular water is not necessarily due to structural change of water such as weaker H-bonding alone, instead reduced vibrational coupling may suppress the 3250 cm^{-1} relative to the 3450 cm^{-1} band. Therefore, a comprehensive discussion of the result should address the issue of vibrational coupling as well.

2. Page 7, equation 2: The thickness of the interfacial water layer is calculated on the basis of eqn 2, in which N is the average number of interfacial water molecules associated with one globular protein. This number (N) can't be determined using MCR analysis, because on the basis of minimum area spectrum in MCR analysis, the SC-spectrum shows only the perturbed water molecules in the hydration shell (NOT all water molecules in the hydration shell). Therefore, at the most, eqn 2 can provide the underestimated thickness of the perturbed water associated with the cellular matrix. According to Eq. (2) the thickness value is 2.6 \AA , which corresponding to just one molecular layer of water all of which are perturbed and does not have any influence on the second layer of water. On reflection, the observed results imply that influence of the globular protein is mostly likely extended beyond the first hydration shell.

The limitation of minimum area spectra in MCR analysis and the associated uncertainty in the thickness may be partly overcome by an alternative analysis called Raman difference spectroscopy with simultaneous curve fitting analysis (Raman-DS-SCF), which provides a more realistic band shape and quantitative information of the OH stretch spectra of the perturbed water (*J. Phys. Chem. C* 2020, 124, 3028–3036).

3. Page 8: "For HeLa cells, the thickness spreads from half layer to nearly two monolayers of water, with an average value of nearly one molecular layer (Fig. 5f). Similar distributions were observed in RAW 264.7 and HT22 cells (Fig. 5g). Hence, we uncover the generality and invariance of biointerfacial water in living cells." This statement needs to be seen in light of the uncertainty or lowest limit of the thickness of the interfacial water as mentioned in the previous comment.

4. In Fig 1b, please show the background spectrum without the live cell i.e. from a position of the Petri-dish without the cell, for same acquisition parameters. Are the spectra shown in Figure 1b represent the experimentally recorded raw spectra or some corrections/normalization are introduced, if so please mention the same.

5. Figure 2c and 2f: As shown in figure 1b, the intensity of the OH stretch signal for the intracellular water is ~ 80 to that of the bulk water. According the intensity of the HOH bend (1640 cm^{-1}) and bend-libration combination (2110 cm^{-1}) bands are also expected to contribute by \sim

80%. However, the spectra shown in Figure 2c and 2f for the hydrated and dehydrated cells have hardly any difference. Why it is so? Also please show the bend-librational combination band (2110 cm^{-1}) as well in Figure 2c and 2f.

6. The results presented in Figure 3a, 3b, 3d, 3g and 3i are difficult to follow; This reviewer fails to understand the shaded figures (3d, 3g and 3i). I guess the authors wanted to show the minimum area non-negative response of the OH stretch band which corresponds to the bulk water contribution of 83% for the cell A. And therefore for bulk water $> 83\%$ the difference SC spectrum becomes negative in the OH stretch region which is unreasonable and for the bulk water contribution $< 83\%$, the SC spectrum is not the minimum area spectrum. Please clarify this point and if possible represent the results in a different format which is easy to follow. Moreover, please show the entire OH stretch region in the X and Y axes in Figure 3b.

7. The perturbed intracellular water is mainly assigned to the water associated with the globular proteins inside the cell. Controlled experiment, such as the spectrum of the hydration shell water of globular protein (in vitro) should be measured to compare with the in vivo results. This comparison will provide further insight into the in vivo and in vitro assignments.

8. Page 4: Based on the proximity probe and nominal spectral shift, it is concluded that the protein conformation change is subtle even after dehydration. Does it mean that hydration of protein is not so crucial for maintain its conformation in live cell? The question is how reliable is the proximity probe for understanding protein conformational change due to dehydration? The consistency of protein conformation needs to be verified by other methods.

9. In the abstract as well as the main text, the authors claim that they have developed a Raman micro-spectroscopy technique to uncover the structural change on water inside a cell. However, it is essentially a modified confocal Raman microscopy technique which has smaller probe volume. Therefore, instead of "developed" it would be more appropriate if it mentions that a modified confocal Raman microscopy has been applied to uncover the intracellular water, and giving emphasis of the modifications. How is the dimension of the illumination spot estimated?

10. How is the MCR analysis performed? The algorithm and the analytical aspects of the MCR need to be discussed in detail in the supporting information.

11. Page 6: "Raman scattering cross section of liquid water is insensitive to hydrogen bonding strength and structure⁴⁴⁻⁴⁶." This is not a statement but an approximation, which should be clarified.

12. Figure 1a: are the two filters LPD1 and LP different? Why the second filter is needed, because the anti-Stokes and the Rayleigh signals are already removed by LPD1?

13. Figure 1a: Mention the beam diameter of the excitation light entering into the 20x objective lens. Also mention the focal length of all the lenses shown in Fig 1a.

14. Figure 1b caption: what is 0.1 MP, is it atmospheric pressure?

15. What type of normalization is used for the comparison of the spectra in Fig 2c?

16. Page 9: ".....an average protein concentration of 1.87 Mm", Mm should be mM.

17. use minor ticks and vertical grids for fig2c,e,f. This will help to compare the changes in the spectra

Reviewer #2:

Remarks to the Author:

This manuscript presents a highly creative approach to measure the characteristics and amount of water that is perturbed by solutes in whole living mammalian cells. The experimental work appears exceedingly careful. On the basis of an extensive analysis of Raman spectromicroscopy of the cells, they report that most of the water in a cell behaves bulklike with only a small amount interacting with biomolecules in the cells. This is presented as surprising and contradicting the standard view of intracellular water so it should be of interest to the broader biological and physical science community.

The technique presented by the authors is marvellous! I love the idea of applying Raman microspectroscopy to disentangle water interacting with cell components from bulk water. It is also

impressive that the authors can generate such high quality spectra (1000:1 SNR) from their instrument. The authors provide extensive information about their method but due to the specialized instrumentation, it is unlikely that others would be able to reproduce these studies.

There are a few problems with the paper that must be addressed before the work can be considered for publication. First, the authors report that they use MCR spectroscopy to tease out the specific response of interfacial or perturbed water from bulk water in a cell, but it is not clear that they actually performed the MCR analysis as pioneered by Ben-Amotz (see Perera, et al, JACS, 2008,130, 4576). It appears that the authors just performed difference spectroscopy to generate the spectrum of "interfacial water". The authors need to replace their simple difference spectroscopy used for the actual MCR method, which enlists a principal component analysis to distinguish the bulk water spectrum from solute perturbed water spectrum (or, if they actually performed the PCR decomposition, then they need to make that very clear).

Second, the authors attribute changes in the water spectrum only in terms of interactions with proteins. But cells have a lot more molecular components that interact with water including lipids (specifically lipid headgroups), carbohydrates, nucleic acids, as well as a bevy of small molecule osmolytes. I recommend that the authors perform a true MCR analysis on various biomolecules and osmolytes to identify the contribution that comes from water interacting with those classes of molecules, then use those results to present a true MCR analysis of the water in the cells.

Finally, although the biology community may embrace a picture where the majority of water in a cell is not bulklike, this is not a particularly surprising result for researchers with more physical views of aqueous systems. It is a little offensive to insinuate (as the authors do on page 9), that their vibrational spectroscopy results are somehow superior to dynamical measures that have shown a large preponderance of bulklike water in cells. Given the 55 M concentration of bulk water, there is a lot of opportunity for water molecules to find each other to interact with and provide a highly bulklike environment even when there are a lot of other materials in close proximity.

The issues raised in this review, at a minimum, require extensive revision and conclusions, and may prohibit publication.

Reviewer #3:

Remarks to the Author:

This manuscript presents the adaption of Raman multivariate curve resolution (MCR) spectroscopy or solvation shell spectroscopy to living cells. The authors find that contrary to common belief, living cells consists mostly of bulk water and only about 3% of biointerfacial water with a significantly altered hydrogen-bonding structure. This corresponds to about one water layer around globular proteins inside the cells. This is in contrast to the normal picture of a crowded cell with mostly biointerfacial water.

This is a very interesting result and a beautiful illustration of what pushing the limits of solvation shell spectroscopy can do. The study is very well performed with all the necessary control experiments. The manuscript is furthermore well written and appropriate for Nature Communications.

I only have a couple of minor comments for consideration but otherwise the manuscript is essentially publishable as is.

Minor comments:

In the introduction it says: "Vibrational sum frequency generation (SFG) has intrinsic interface selectivity but requires an extended planar interface^{30,31}". This is strictly speaking not correct.

Chirality makes SFG allowed in the bulk and the chirality transfer from biomolecule to the surrounding solvent makes it possible to probe biointerfacial water directly, being the water directly bound to the biomolecule. Actually, later in the manuscript, a chiral SFG study from Elsa Yan and coworkers (reference 70) is described, which illustrates exactly this. However, this study was predated by 2 studies of Petersen and coworkers: <https://doi.org/10.1021/acscentsci.7b00100> and <https://www.science.org/doi/full/10.1126/sciadv.aao5603>. Accordingly, chiral SFG can indeed be used to probe biointerfacial water through this chirality transfer in a label-free way. Even though these examples study biomolecules attached to flat interfaces, chiral SFG does not require a flat surface. However, this chirality transfer is indeed very short range since it relies on the chirality transfer from biomolecule to solvent, as the authors discuss in connection with reference 70.

On page 5 of the manuscript it reads: "Raman-MCR has been primarily used in small molecules such as alcohols and ions with relatively large hydration shell and hence negligible solute contribution³⁴."

This is correct for Raman-MCR, but I would like to point out that the MCR method has previously been applied to biomolecules in the form of FTIR-MCR: <https://doi.org/10.1021/acs.jpcllett.6b02925>

Editorial Note: In the response to reviewers, the authors refer to Fig. S15 which is no longer included in the manuscript, but remains in the Peer Review file for reference.

Response to Reviewer 1:

Water intricately regulates biological processes, yet the structure of intracellular water remains enigmatic. This study addresses critical questions about differences in the hydrogen-bonded network within living cells compared to bulk water, exploring its abundance, location, and new structural features. To overcome the experimental challenges, the authors used confocal Raman microscopy and MCR analysis along with the reference Raman spectrum of vacuum dehydrated cell components. The results suggest a consistent (~3%) population of structurally-altered non-bulk-like water near biomolecules. It exposes a weakened hydrogen-bonded network and heightened disorder. Statistical constancy across cells could be an indication of a universal biophysical role of such interfacial water, marking this research as a gateway to in vivo water structure exploration. The paper would be of interest to the readership of the journal, Nature and may be accepted for publication and careful consideration of the following concerns.

Response: We sincerely appreciate reviewer 1's thoughtful analysis and recognition of our study's contribution to understanding intracellular water and its role in biological processes. We have performed new experiments and addressed the technical issues in our revised manuscript.

1. The blue shift of the central frequency of the OH stretch band is taken as evidence of weaker H-bonding of the perturbed intracellular water. However, the analysis of the OH stretch band of water is not so straightforward due to the presence of intra and intermolecular coupling. In fact, the 3250 cm⁻¹ band of the OH stretch spectrum is not solely due to strongly H-bonded water, but mainly due to collective nature of the OH-stretch vibration of water. The collective vibration of water is affected at interface and in the hydration shell of a solute or macromolecule and it may either increase or decrease depending on the nature of the solute (J. Chem. Phys. 128, 224511 2008; Phys. Chem. Chem. Phys., 2010, 12, 982–991; J. Mol. Liq. 390 (2023) 122987; J. Phys. Chem. B 2013, 117, 16479–16485). Therefore, the spectral change of the intercellular water is not necessarily due to structural change of water such as weaker H-bonding alone, instead reduced vibrational coupling may suppress the 3250 cm⁻¹ relative to the 3450 cm⁻¹ band. Therefore, a comprehensive discussion of the result should address the issue of vibrational coupling as well.

Response: We thank reviewer's insightful comment regarding the interpretation of the blue shift of O-H stretch average frequency. We agree that recent literature has attributed this 3250 cm⁻¹ peak's fundamental origin to Fermi resonance between the OH-stretch modes and the bending overtone and attributed the link between this shoulder and tetrahedral order to the fact that strong hydrogen bonds are required to the redshift the OH-stretch frequencies down to the bending overtone.

Following the reviewer's suggestion, in our revised manuscript, we have expanded our discussion to consider the potential impact of reduced vibrational coupling on the observed spectral features:

One may take a closer look when interpreting the molecular basis of the spectral changes of O-H stretch^{1, 2}. It is noted that the observed suppression of the 3250 cm⁻¹ band relative to the 3450 cm⁻¹ band might not only signify a transformation towards a weaker hydrogen-bonding network in the vicinity of biomolecules but could also reflect reduction in the collective vibrational coupling within the water matrix. Such change lessens the delocalization of vibrational energy across multiple water molecules, potentially leading to decreased spectral response at 3250 cm⁻¹ and a blue shift in the central frequency of the OH stretch compared to bulk water. Isotopic dilution techniques, which have been effective in differentiating between the impacts of hydrogen bonding perturbation and vibration coupling in studies of ion-induced and hydrophobic hydration shells, face considerable challenges in a cellular context^{3, 4}. Despite the complexity of molecular narrative of spectral change, both the weakening of hydrogen bonds and the reduction in vibrational coupling converge on a consistent structural theme: a more disordered structure in biointerfacial water.

2. Page 7, equation 2: The thickness of the interfacial water layer is calculated on the basis of eqn 2, in which N is the average number of interfacial water molecules associated with one globular protein. This number (N) can't be determined using MCR analysis, because on the basis of minimum area spectrum in MCR analysis, the SC-spectrum shows only the perturbed water molecules in the hydration shell (NOT all water molecules in the hydration shell). Therefore, at the most, eqn 2 can provide the underestimated thickness of the perturbed water associated with the cellular matrix. According to Eq. (2) the thickness value is 2.6 Å, which corresponding to just one molecular layer of water all of which are perturbed and does not have any influence on the second layer of water. On reflection, the observed results imply that influence of the globular protein is mostly likely extended beyond the first hydration shell.

The limitation of minimum area spectra in MCR analysis and the associated uncertainty in the thickness may be partly overcome by an alternative analysis called Raman difference spectroscopy with simultaneous curve fitting analysis (Raman-DS-SCF), which provides a more realistic band shape and quantitative information of the OH stretch spectra of the perturbed water (J. Phys. Chem. C 2020, 124, 3028–3036).

Response: We thank reviewer for the insightful comments on the limitation of Raman-MCR, particularly concerning the “rotational ambiguity” in the spectra derived from MCR analysis for quantitative assessments.

As the reviewer rightly pointed out, the “minimum area” SC spectrum essentially defines the tightest solvation-shell boundary, while the other members of the family of rotation ambiguity spectra pertain to extending the boundary farther out from the solute⁵. This implies that the minimum-area SC spectrum does not necessarily encompass all water molecules in the first hydration shell, but rather signifies the “most perturbed water”. Also note that our study has adopted an upgraded version of Raman-MCR without any normalization, to retain the intensity of the solute-affected water spectrum for retrieving the number of water molecules affected by biomolecules. It is also crucial to recognize the features identified in “minimum area” SC

spectrum might originate from either a few highly perturbed solvent molecules or from a larger number of less perturbed molecules. This perspective aligns with the reviewer's comment that "eqn 2 can provide the underestimated thickness of the perturbed water associated with the cellular matrix".

The spectroscopic rotational-ambiguity may be resolved, by leveraging an experimental or simulation-based estimate of the number of water molecules in the first solvation-shell spectrum⁵. It is important to note that recent computational studies provide valuable insights into short-range hydration dynamics around proteins, suggesting the confinement of significant hydration effects to the first hydration shell. This computational evidence bolsters our experimental approach to quantifying biointerfacial water as effectively constituting one complete layer of water molecules hydrating intracellular proteins. In our manuscript, we state the following:

Eq.(2) yields a thickness value of 2.6 Å, corresponding to just one molecular layer of water. Importantly, a view of short-range protein hydration is emerging in the literature^{55,67-70}. In particular, a systematic computational study of the spatial range of protein hydration reveals all investigated structural and dynamical properties have exponential decay lengths of less than one hydration shell⁵⁵. Since rotational ambiguity renders Raman-MCR algorithm to give the lower bound for the abundance of biointerfacial water⁴⁴, the combination with the upper bound of just the first hydration shell (strongly suggested by the computational study⁵⁵) encourages us to quantify the biointerfacial water as one full water layer hydrating intracellular proteins"

Furthermore, we acknowledge the relevance of alternative analytical methods like Raman-DS-SCF, as suggested by reviewer. This method's potential is discussed comparatively in Jahur Alam Mondal's review article "*Vibrational Raman Spectroscopy of the Hydration Shell of Ions*"⁶. Comparisons within specific contexts, such as the hydration shell of low-charge-density anions, show similar band shapes between spectra extracted via Raman-MCR and Raman-DS-SCF. Nonetheless, a distinct signature, such as a band around 3600 cm⁻¹ observed in MCR-extracted spectra for Mg²⁺, due to the minimum area subtraction employed in MCR. In contrast, DS-SCF tends to uncover more subtle features of water, but requires rational choice of fitting function. It is important to note that both Raman-MCR and Raman-DS-SCF are inherently biased towards the perturbed water in the hydration shell⁶. Consequently, the hydration water component having bulk-like vibrational response is generally not retained in the extracted spectrum.

These considerations inform our methodological choices, balancing the trade-offs between different analytical approaches based on the solvation system under study. To comprehensively compare these, we have included new Supplemental Discussion 5 and Figure S8 in our revised manuscript. Again, our method represents an adaption and expansion of the original two-component Raman-MCR technique, specifically tailored for the context of live cells. Herein we highlight different methods in the context below.

Raman-MCR spectroscopies^{5, 7} and Raman difference Spectroscopy with simultaneously curve fitting (**Raman-DS-SCF**)⁶ are essentially differential spectroscopy techniques, coupled with different optimization criteria. They both are used to

quantitatively retrieve the spectra of water perturbed by ions/solutes (termed as solvation/hydration shell). In both analyses, the solution spectrum is considered as a linear combination of three spectral components: (1) intrinsic vibration response of the solute/ion (if any). (2) spectrum of water that is perturbed by the ion/solute. (3) spectrum of unperturbed water that is equivalent to the spectrum of the bulk water. Component (1) and (2) are combinedly treated as solute-correlated (SC) spectrum or the hydration/solvation shell spectrum. To retrieve SC information, the question becomes how much bulk water spectrum [reference spectrum, $R(\tilde{\nu})$] shall be subtracted from experimentally measured solution spectrum ($S(\tilde{\nu})$).

$$SC(\tilde{\nu}) = S(\tilde{\nu}) - f * R(\tilde{\nu})$$

Raman-DS-SCF⁸ determines “optimal fraction” of f by simultaneously fitting $S(\tilde{\nu})$ with four Gaussian bands and reference spectrum $R(\tilde{\nu})$.

$$SC(\tilde{\nu}) = S(\tilde{\nu}) - f * R(\tilde{\nu}) = \sum_{n=1}^n \left(A_n e^{-\frac{(\tilde{\nu} - \tilde{\nu}_n)^2}{2\Gamma_n^2}} \right) + A_0$$

Given the hydration shell water is basically “perturbed bulk water”, the number of Gaussian components and the initial guess of coefficients are guided by four component Gaussian fitting of bulk water spectrum ($\tilde{\nu}_1 \approx 3000 \text{ cm}^{-1}$, $\tilde{\nu}_2 \approx 3250 \text{ cm}^{-1}$, $\tilde{\nu}_3 \approx 3450 \text{ cm}^{-1}$, $\tilde{\nu}_4 \approx 3600 \text{ cm}^{-1}$), recorded at the same experimental conditions as that of solution spectrum. To get physically meaningful band shape, RD-SCF analysis applies restrictions, such as non-negative coefficients, and the non-negative spectrum during fitting optimization. A critical aspect of validation is the reproducibility of $SC(\tilde{\nu})$ extracted by this method. Not only should this spectrum be highly reproducible, but the quantitative information it yields should also be in excellent agreement with findings from X-ray diffraction and Molecular Dynamics (MD) simulation studies.

Two-component Raman-MCR analysis determines solute-affected water spectra $SC(\tilde{\nu})$ based on area-normalized minimum area criterion. It involves the systematic subtraction of maximum fraction (f) of $R(\tilde{\nu})$ from $S(\tilde{\nu})$. The subtraction is continued until there appears a *spectral point reaches zero signal* within the vibration resonance of O-H stretch ($\sim 3000 - 3700 \text{ cm}^{-1}$). This makes Raman-MCR free from band-shape approximation, especially the experimentalist’s choice of fitting function. Compared to Raman-MCR, RD-SCF is free from constrains of area normalization and minimization but needs input fitting function (retain the intensity and band-shape).

$$SC(\tilde{\nu}) = S(\tilde{\nu}) - f_{max} * R(\tilde{\nu}) \rightarrow \textit{nonnegative minimum area}$$

Our approach represents an adaption and expansion of the original two-component Raman-MCR technique, specifically tailored for the context of live cells. In our model, the water spectra within live cells are considered as the linear combination of three distinct components:

1. Intrinsic vibration response of the solute, corresponding to the dehydrated state of cell.
2. Spectrum of significantly perturbed water. We refer to this as bio-interfacial water, highlighting its altered spectral feature due to close interactions with cellular components.
3. Spectrum of unperturbed water that is equivalent to the spectrum of the bulk water.

$$BioWater(\tilde{\nu}) = LiveCell(\tilde{\nu}) - x * Ref(\tilde{\nu}) - y * DehydrCell(\tilde{\nu}) \rightarrow \text{Non-negative Minimum Area}$$

$$y = \frac{LiveCell_{2930} - x * Ref_{2930}}{DehydrCell_{2930}} \rightarrow \text{Mass Conservation Restriction (at } 2930 \text{ cm}^{-1}\text{)}$$

In our approach, we apply simultaneous component fitting during subtraction optimization and visualize the differential results in the MCR residual map. We adapted the core principles of MCR analysis to isolate the vibrational responses of “most perturbed water” by computing the non-negative minimum area of the spectrum. However, we recognized that merely achieving a single point of zero in the spectrum is not always sufficient or indicative of optimal results. Thus, it is remarkably significant in our algorithm because at that particular point, the difference spectrum from 3054 cm^{-1} to 3150 cm^{-1} simultaneously reaches the local minimum. This collective behavior not only guarantees the non-negative minimum-area condition to be fulfilled, as emphasized in Raman-MCR; but serves as an internal check to suggest the existence of an intrinsic spectrum. Subsequently, the bio-interfacial water spectrum is fitted with a minimum number of Gaussian or product of Gaussian to resolve the spectral components. In our analysis, the “differential” and “curve fitting” analyses were carried out sequentially.

Fig. S8. Overview of Raman-MCR analysis workflow. Input: single cell Raman spectrum and bulk water Raman spectrum at same acquisition condition. Output: MCR residual map and difference spectra (bio-interfacial water spectra). The process involves two main steps: (1) *Solvent contribution removal*: A variable fraction ($x\%$) of bulk water spectrum is subtracted from the single cell spectrum to produce Solute-correlated (SC) spectra as a function of (x). (2) *Ensemble solute contribution removal*: Dehydrated cell spectra are normalized based on CH band intensity (2930 cm^{-1}) within SC spectra. These normalized spectra are then subtracted from SC spectra to produce difference spectra for each x . The intensity within difference spectra is divided into three categories: positive, negative, near-zero (below a specified threshold), which are used to construct the MCR residual map for each x value. A critical point is identified at the value of x ($x = x_2$) where the minimum non-negative area is observed.

3. Page 8: “For HeLa cells, the thickness spreads from half layer to nearly two monolayers of water, with an average value of nearly one molecular layer (Fig. 5f). Similar distributions were observed in RAW 264.7 and HT22 cells (Fig. 5g). Hence, we uncover the generality and invariance of biointerfacial water in living cells.” This statement needs to be seen in light of the uncertainty or lowest limit of the thickness of the interfacial water as mentioned in the previous comment.

Response: In light of the reviewer's previous comment regarding the potential underestimation of the interfacial water layer thickness due to the limitations of the "minimum area" criterion in MCR analysis, we recognize the need for a cautious interpretation of these findings. The reported thickness values, should indeed be considered in the context of this methodological constraint, which might bias the analysis towards the most perturbed water molecules and not fully account for all molecules in the hydration shell. (*See our response to comments #2 regarding rotational ambiguity above*)

To address this point more explicitly, we have revised manuscript to acknowledge the potential underestimation of the hydration layer thickness due to the analytical approach employed. We also emphasize that despite this limitation, the consistency of our findings across different cell types underscores the robustness of the observed trend in biointerfacial water properties. Moreover, we highlight the importance of integrating these results with complementary analytical techniques and theoretical insights to build a more comprehensive understanding of hydration dynamics at the cellular interface.

By doing so, we aim to provide a balanced view of our findings, acknowledging the inherent limitations of the method while also drawing attention to the broader implications and generality of the observed biointerfacial water properties in living cells.

4. In Fig 1b, please show the background spectrum without the live cell i.e. from a position of the Petri-dish without the cell, for same acquisition parameters. Are the spectra shown in Figure 1b represent the experimentally recorded raw spectra or some corrections/normalizations are introduced, if so please mention the same.

Response: We thank the reviewer for pointing out this issue. Following the reviewer's suggestion, in the revision, we have incorporated the background spectrum from a blank region of the petri dish (indicated by the gray line in Fig. 1b), which was obtained under identical conditions (same power and integration time). The spectral data presented in Fig. 1b reflects the raw experimental recordings, post-baseline correction. This correction involved subtracting the background intensity observed at 3600 cm^{-1} . We did not apply any further corrections or normalization to this data. An explanation of this process has been included in the *Supplemental Information* for clarity:

Baseline correction in spectra data: The baseline correction was performed by subtracting the background intensity at 3600 cm^{-1} , without applying further corrections or normalization.

5. Figure 2c and 2f: As shown in figure 1b, the intensity of the OH stretch signal for the intracellular water is ~ 80 to that of the bulk water. According to the intensity of the HOH bend (1640 cm^{-1}) and bend-libration combination (2110 cm^{-1}) bands are also expected to contribute by $\sim 80\%$. However, the spectra shown in Figure 2c and 2f for the hydrated and dehydrated cells have hardly any difference. Why is it so?

Response: We thank reviewer for bringing up this discussion. In Fig. 2c, we focus on discerning structural changes by examining how amide Raman spectral features alter during dehydration. A key step in our analysis involves the removal of water interference from single-cell spectra. This is crucial because the H-O-H bend peak at 1640 cm^{-1} overlaps with the broad Amide I band, complicating the analysis. By eliminating this background, we enable a more precise and direct comparison of the Amide I and III bands as they undergo dehydration. Essentially, the single cell spectra depicted in Fig. 2c represent what we term as solute-correlated spectra (SC-spectra). This explains the minimal variance observed between the single-cell spectra (Blue/Yellow line) and dry cell spectra (Red line). To avoid confusion, we have revised the corresponding legend labels and figure caption to better reflect the data analysis conducted, indicating single cell spectra with water background removed.

To offer a comprehensive perspective, we have included Fig. R1 below. This figure juxtaposes the single-cell spectra (without water background removal) against those presented in Fig. 2c. In this comparison, the difference between the single-cell spectra, represented by blue dashed and yellow dashed lines, and the dry cell spectra, indicated by a red dashed line, becomes evident, highlighting the contributions from intracellular water of HOH bend (1640 cm^{-1}) and bend-libration combination band (2110 cm^{-1}).

Fig. R1: Comparison between Spectra with water BG removal and Original Spectra

In Fig. 2f, similar data processing is applied on BSA solution to remove the pure water background. As a result, the blue curve in this figure effectively represents a solute-correlated spectrum. To ensure clarity, we have updated the legend labels accordingly.

Also please show the bend-librational combination band (2110 cm^{-1}) as well in Figure 2c and 2f.

Response: We have updated Fig. 2c and 2f to cover 2110 cm^{-1} band within Raman spectral window in the revised manuscript.

6. The results presented in Figure 3a, 3b, 3d, 3g and 3i are difficult to follow; This reviewer fails to understand the shaded figures (3d, 3g and 3i). I guess the authors wanted to show the minimum area non-negative response of the OH stretch band which corresponds to the bulk water contribution of 83% for the cell A. And therefore, for bulk water $> 83\%$ the difference SC spectrum becomes negative in the OH stretch region which is unreasonable and for the bulk water contribution $< 83\%$, the SC spectrum is not the minimum area spectrum. Please clarify this point and if possible, represent the results in a different format which is easy to follow.

Response: We thank reviewer for bringing up this issue. In the revision we have added a new Fig. S8 to elaborate on the Raman-MCR analysis and the shaded figures (3d, 3g and 3i). To enhance the clarity, we modified the Fig. 3 caption to highlight the points of interpretation the critical point.

Moreover, please show the entire OH stretch region in the X and Y axes in Figure 3b.

Response: We have updated Fig. 3b to show the entire OH Region in the revised manuscript.

7. The perturbed intracellular water is mainly assigned to the water associated with the globular proteins inside the cell. Controlled experiment, such as the spectrum of the hydration shell water of globular protein (in vitro) should be measured to compare with the in vivo results. This comparison will provide further insight into the in vivo and in vitro assignments.

Response: We thank reviewer for this suggestion. In our revision we have newly added Fig. S15 to include the suggested control experiments on globular protein hydration shell measurement. We selected two globular proteins, Bovine Serum Albumin (BSA) and lysozyme, and prepared their aqueous solutions at concentrations ranging from 50 – 150 mg/ml. This concentration range was carefully chosen to prevent protein aggregation. We also obtained dehydrated protein spectra from dry films of these proteins, made through vacuum dehydration over two days. Employing similar Raman-MCR analysis as in our cell studies, we extract the Raman spectra of protein hydration shell at various concentration (Fig. S15e). Qualitatively, the hydration shell of BSA and lysozyme measured *in vitro* closely mirror the spectral shape of bio-interfacial water observed in single cells (example indicated by the green line). The high resemblance is characterized by two prominent peaks and displayed a more disordered structure ($A_{3290}/A_{3460} < 1$). However, the *in vitro* spectra exhibit narrower peak widths compared to cellular spectra, likely due to the averaging effects across diverse proteins within cells, suggesting a more complex interaction landscape *in vivo*. These findings, while showing some heterogeneity, provide valuable comparative insights and bolster our hypothesis regarding the dominant influence of protein interactions on cellular water spectra.

Fig. S15. In-vitro experiments on globular protein hydration shell. (a-d) Raman-MCR analysis on BSA (50 mg/ml in DI water) and Lysozyme (50 -150 mg/ml in DI water) solutions. The BSA and Lysozyme Raman spectra are obtained at 50 mW over 300 s. **(e)** Comparison between the *in vivo* bio-interfacial water spectra from single cells and the *in vitro* extracted hydration shell spectra of BSA and Lysozyme.

8. Page 4: Based on the proximity probe and nominal spectral shift, it is concluded that the protein conformation change is subtle even after dehydration. Does it mean that hydration of protein is not so crucial for maintain its conformation in live cell? The question is how reliable is the proximity probe for understanding protein conformational change due to dehydration? The consistency of protein conformation needs to re-verified by other methods.

Response: We appreciate the reviewer's question regarding our hypothesis "subtle protein conformation changes upon vacuum dehydration", as inferred by the minimal spectral shifts of amide I and III bands. To address the reviewer's concern, we summarize the hypotheses and evidence chain in figure below.

Fig. R2. Hypotheses summarized from subtle spectral shifts in amide I & III

It is important to note that **Hypothesis #2** is drawn from a specific measurement in our study. The spectral analysis, while indicative, is not exhaustive in fully capturing the intricate complexity of protein conformational changes. The reliability of Raman probe in detecting conformational changes (upon dehydration) may be influenced by factors such as the sensitivity of the spectral shifts to conformational alterations. **In light of it, we agree with reviewer that our implication of protein conformation should be further substantiated using additional methods,** such as X-ray crystallography, NMR, and CD spectroscopy. Spectroscopy-wise, to further prove the hypothesis, we conducted a comparative analysis between the spectral shifts observed during vacuum **dehydration** and those resulting from protein **denaturation**, the latter of which is known to induce irreversible conformational changes (Fig. S5). Notably, denaturation led to significant alterations in the spectral shape (particularly in the Amide III region) and spectral shifts in the Amide I region that were over sevenfold larger than those observed during vacuum dehydration (1.4 cm⁻¹ v.s. 9.8 cm⁻¹). This comparison supports the conclusion that isothermal

vacuum dehydration induces only modest perturbations to the spectral profile and, by extension, to protein conformation. It is important to note that **the subtle spectral change of amide bands during dehydration does not diminish the overall importance of hydration in protein conformation**; but indicates that the changes were not pronounced enough to significantly alter the spectral signals we analyzed.

Despite these concerns, the proximity probe in our study still provides valuable insights in supporting **Hypothesis #1**, which is of more relevance to the MCR analysis. Quantitatively, the estimated spectral shift for N-H due to dehydration is less than 4 cm⁻¹. To evaluate the robustness of our Raman-MCR analysis against such spectral variations, we conducted a simulation using artificially shifted dehydrated spectra (Fig. S9). This exercise aimed to ascertain the extent to which deviations in the N-H region could influence our analysis. The results of this simulation revealed that our Raman-MCR analysis possesses a higher degree of tolerance for shifts in the N-H region than for alterations in the reference water spectra. This finding suggests that, even in the presence of considerable spectral changes within the N-H region (upon dehydration), the overall integrity and conclusions of our Raman-MCR analysis remain robust and reliable.

9. In the abstract as well as the main text, the authors claim that they have developed a Raman micro-spectroscopy technique to uncover the structural change on water inside a cell. However, it is essentially a modified confocal Raman microscopy technique which has smaller probe volume. Therefore, instead of “developed” it would be more appropriate if it mentions that a modified confocal Raman microscopy has been applied to uncover the intracellular water, and giving emphasis of the modifications. How is the dimension of the illumination spot estimated?

Response: We thank reviewer for the suggestion and we have modified the terminology used in our manuscript regarding our Raman micro-spectroscopy technique.

We also added a more detailed description on the estimation of illumination spots in *Supplemental Information (SI)*.

The diffraction-limited spot sizes, both laterally and axially, are calculated using a simplified equation derived from the more complex Airy disk formula. In our experimental setup, with a wavelength (λ) of 532 nm, and an effective numerical aperture (N.A.) ranging from 0.3 to 0.4, determined by the beam's dimensions at the objective lens's back aperture (20x, air), we estimated the confocal volume to be approximately 2 μm x 2 μm x 10 μm .

$$d_{xy} = \frac{1.22 * \lambda}{N.A.} = [1.6 \sim 2.2] \mu\text{m}$$

$$Z = \frac{2 * n * \lambda}{(N.A.)^2} = [6.65 \sim 11.8] \mu\text{m}$$

10. How is the MCR analysis performed? The algorithm and the analytical aspects of the MCR need to be discussed in detail in the supporting information.

Response: We appreciate the reviewer highlighting this matter. To address it, we have expanded our revised manuscript to include new *Supplemental Discussion 5* and Figure S8, which provide a detailed explanation of the MCR algorithm and its analytical considerations.

11. Page 6: “Raman scattering cross section of liquid water is insensitive to hydrogen bonding strength and structure⁴⁴⁻⁴⁶.” This is not a statement but an approximation, which should be clarified.

Response: We have updated the manuscript to highlight this point:

Raman scattering cross section of liquid water is generally considered relatively insensitive to changes in hydrogen bonding strength and structure.

12. Figure 1a: are the two filters LPD1 and LP different? Why the second filter is needed, because the anti-Stokes and the Rayleigh signals are already removed by LPD1?

Response: We thank reviewer for the question. The LPD1 (LPD02-532RU-25, Semrock, OD = 3 @ 532nm) and LP (LP03-532RU-25, Semrock, OD = 7 @ 532 nm) are both used to block 532 nm light. Even though the LPD1 removes significant portion of anti-Stokes and Rayleigh signals, some of unwanted signals might still pass through due to imperfect rejection of dichroic filters from our experience. The additional filtering with LP (with higher OD) ensures the detected signal is predominantly from Raman scattering for accurate analysis.

13. Figure 1a: Mention the beam diameter of the excitation light entering into the 20x objective lens. Also mention the focal length of all the lenses shown in Fig. 1a.

Response: We thank reviewer for the questions. We have added the beam size and focal length details in the Fig. 1a caption.

The beam diameter of the excitation light entering into 20x objective lens is 9-10 mm. The focal length for lenses: L1/L2(f= 40/300 mm), L3/L4(f=150/50 mm), L5/L6(f=100/100 mm).

14. Figure 1b caption: what is 0.1 MP, is it atmospheric pressure?

Response: We would like to thank the reviewer for the question. 1atm = 0.1 MPa, representing atmospheric pressure. We have noted in the Fig. 1b caption for better clarity.

15. What type of normalization is used for the comparison of the spectra in Fig 2c?

Response: The single cell spectra (blue and yellow line) shown in Fig. 2c are not normalized. The dehydration spectrum (red dashed line) is normalized to blue line on CH₃ peak at 2930 cm⁻¹. Below we shown the full range window:

Fig. R3. Expanded view of Fig. 2c with complete spectral range.

16. Page 9: “.....an average protein concentration of 1.87 Mm”, Mm should be mM.

Response: We would like to thank the reviewer for pointing out this typo. We have corrected it in the manuscript.

17. Use minor ticks and vertical grids for fig2c,e,f. This will help to compare the changes in the spectra

Response: We would like to thank the reviewer for the suggestion. We have updated the Fig. 2c,e,f with minor ticks to help comparison.

Response to Reviewer 2:

This manuscript presents a highly creative approach to measure the characteristics and amount of water that is perturbed by solutes in whole living mammalian cells. The experimental work appears exceedingly careful. On the basis of an extensive analysis of Raman Spectro-microscopy of the cells, they report that most of the water in a cell behaves bulklike with only a small amount interacting with biomolecules in the cells. This is presented as surprising and contradicting the standard view of intracellular water so it should be of interest to the broader biological and physical science community.

The technique presented by the authors is marvelous! I love the idea of applying Raman micro-spectroscopy to disentangle water interacting with cell components from bulk water. It is also impressive that the authors can generate such high-quality spectra (1000:1 SNR) from their instrument. The authors provide extensive information about their method but due to the specialized instrumentation, it is unlikely that others would be able to reproduce these studies.

We sincerely thank the reviewer for the support of our work and for the suggestions. We have performed new experiments and addressed the individual issues as follows:

There are a few problems with the paper that must be addressed before the work can be considered for publication.

1. First, the authors report that they use MCR spectroscopy to tease out the specific response of interfacial or perturbed water from bulk water in a cell, but it is not clear that they actually performed the MCR analysis as pioneered by Ben-Amotz (see Perera, et al, JACS, 2008,130, 4576). It appears that the authors just performed difference spectroscopy to generate the spectrum of “interfacial water”. The authors need to replace their simple difference spectroscopy used for the actual MCR method, which enlists a principal component analysis to distinguish the bulk water spectrum from solute perturbed water spectrum (or, if they actually performed the PCR decomposition, then they need to make that very clear).

Response: We appreciate the reviewer’s comments regarding our methodology. To address the concerns raised, we have expanded the description and provided a comparative discussion in the newly added **Supplemental Discussion 5** to clarify our methods.

First of all, we would like to refer to the discussion on vibrational-MCR spectroscopy to Dor Ben-Amotz’s review article “Hydration-shell vibrational Spectroscopy”⁵ to elucidate the technical nuances.

“Vibrational-MCR is essentially a difference spectroscopy that may be conveniently implemented using a matrix algebraic algorithm called self-modeling curve resolution (SMCR). The resulting SMCR decomposition of the measured vibrational spectra of the pure solvent and solutions containing a solute of interest yields a solute-correlated

spectrum revealing vibrational features arising both from the solute itself and from any solvent molecules that are perturbed by the solute. In other words, a SC spectrum is equivalent to the non-negative minimum area difference between the solution and pure solvent spectra. ... Although vibrational-MCR is most conveniently implemented using SCMR, other methods (including manual direct subtraction) may be used to obtain essentially identical SC spectra.”

Hence, SMCR involves Singular Value Decomposition (SVD)/Principal Component Analysis (PCA) and decomposes the spectra into two physically relevant component – one representing the bulk solvent, and the other component depicting the non-negative minimum area solute-corrected (SC) spectrum. This process aligns methodologically with direct differential analysis, provided it strictly adheres to the non-negative minimum area criterion. To demonstrate, we performed SMCR decomposition and manual subtraction (with minimum area constraints) on small molecules like TBA, yielding equivalent hydration shell spectra results (red and yellow line in Fig. R4). Simply speaking, SMCR is a special, mathematically rigorous sub-category of difference spectroscopy.

Fig. R4. TBA Solute-correlated spectra

Furthermore, given that SMCR traditionally limits itself to two-component spectral decompositions, we have adapted this principle for the more intricate live cell context, which involves three components. While PCA is mathematically robust, it lacks physical transparency, which is why we opted for a physically rigorous approach. Our method hinges on subtraction optimization that is deeply grounded on physically justified spectral responses. We evolved from the basic criterion of “a single spectral point reaching zero” to a more rigorous and comprehensive approach, where a “small spectral window collectively reaches zero”, to fulfill the minimum area requirement for the hydration shell spectra. This adaptation is not a mere simplification but a careful, methodical enhancement of differential spectroscopy. It underscores our understanding and meticulous application of these techniques in a biologically complex

setting, ensuring both scientific rigor and relevance. These important advances are detailed in the newly added ***Supplemental Discussion 5***.

2. Second, the authors attribute changes in the water spectrum only in terms of interactions with proteins. But cells have a lot more molecular components that interact with water including lipids (specifically lipid headgroups), carbohydrates, nucleic acids, as well as a bevy of small molecule osmolytes. I recommend that the authors perform a true MCR analysis on various biomolecules and osmolytes to identify the contribution that comes from water interacting with those classes of molecules, then use those results to present a true MCR analysis of the water in the cells.

Response: We sincerely thank the reviewer for the insightful comment on the molecular origin of the biointerfacial water. We acknowledge the point about the presence of other molecular components in cells, such as lipids, carbohydrates, nucleic acids, and various osmolytes, also interacting with water. Such solute background and its perturbation on water can be intrinsically complicated in living cells. Attempting to separately analyze each component using MCR and then integrate them to understand the collective behavior of water in the cellular context presents significant challenges.

Instead, we argue that it is reasonable and more productive to consider the collective contribution of solutes in MCR analysis and then deduce the predominant molecular factors influencing the hydration spectrum. In our study, we attributed bio-interfacial water primarily due to interactions with proteins, based on our models and supporting evidence from literature that aligns with our results. To address concerns regarding the contribution of these other biomolecules, we present additional experimental evidence as below. Together, they bolster our findings and conclusions.

Experimental Evidence 1: Consistency from spatially-averaged spectra in despite of solute composition variations.

Our approach utilized spatially-averaged spectra, which was acquired with a low N.A. objective lens to reduce cell-to-cell variations. Fig. R5 shows additional examples of single cell spectra and their corresponding biointerfacial water spectra. Despite efforts to standardize the measurement, variations in solute composition were still observable in certain cases, as indicated by differing intensity ratios between the 2853 cm^{-1} and 2930 cm^{-1} peaks. These variations, particularly concerning lipid compositions, had the potential to introduce variability into the water spectra through unique solute-water interactions, **assuming a significant role of lipids**. To mitigate the potential impact of such variations, we meticulously selected dry cell spectra that closely matched the solute background, particularly in terms of their CH spectral profiles. This careful selection process enabled us to successfully isolate and eliminate the influence of the solute background from MCR analysis. The resulting consistency in the biointerfacial water spectra is remarkable. It strongly indicates that the observed spectral signature of biointerfacial water is primarily shaped by consistent factors present across all cell, albeit the variability introduced by lipid compositions.

Fig. R5. Raman-MCR analysis on single cell spectra with varied cellular compositions. (a)-(c) The decreasing intensity ratio between 2853 cm^{-1} and 2930 cm^{-1} peaks indicates reduced lipid contents.

Experimental Evidence 2: Consistency from spatial-resolved spectra across subcellular locations.

In this section, we transition from our initial whole-cell spectroscopy approach to a more focused analysis of subcellular regions. Fig. R6 presents the common stimulated Raman scattering (SRS) imaging of live cells, highlighting the distinct intracellular distribution of various biomolecules. Notably, lipid membranes are predominantly located outside the nucleus, while DNA/RNA are confined within it. In contrast, protein distribution appears more homogenous throughout the cell. **This distribution raises the hypothesis that if protein were not the primary contributor to the observed biointerfacial water spectra, we would have observed spatial variation in the spectra, correlating with the uneven distribution of lipids and DNA/RNA, especially inside and outside the nucleus.**

Fig. R6. Example of multi-color SRS microscopy on living tumor cells: (a) DNA (magenta) at 2967 cm^{-1} , proteins (blue) at 2926 cm^{-1} and lipids (green) at 2850 cm^{-1} . (b) Raman spectrum extracted from

the cell pellet showing the signatures of the different species. Scale bar = 10 μm . Figures adapted from (Lu et al., 2015 PNAS)

To test this hypothesis, we utilized a 60x objective lens (effective N.A. = 1.0) to target smaller subcellular regions for live cell spectra collections (Fig. R7a) and performed subsequent Raman-MCR analysis on spectra from various subcellular locations. In Fig. R7 b-c, we present bio-interfacial water spectra retrieved from subcellular areas both inside and outside the nucleus. Our findings reveal that the water spectra from these different locations are qualitatively similar, characterized by two prominent peaks around 3280 cm^{-1} and 3450 cm^{-1} . This consistency was observed across 52 nucleus regions and 42 non-nucleus regions in 22 HeLa cells (Fig. R7d-e), suggesting a relatively uniform water spectral profile throughout the cell. The similarity in the spectral shape of bio-interfacial water, irrespective of the cellular location, indicates a semi-uniform water spectral profile throughout the cell. **This finding suggests that while other biomolecules may interact with water, their contribution to the water spectral features is not predominant compared to proteins.** Essentially, the biointerfacial water spectral features we identified remain largely unaffected by the variability in other solutes (lipids, DNA/RNA, etc.). This reinforces our conclusion regarding the primary role of proteins.

Fig. R7. Single cell Raman spectroscopy with 60x objective lens. (a) Bright field image of HeLa cells under a 60x objective lens. (b-c) Raman-MCR analysis on spectra obtained from specific subcellular locations, as indicated by the white arrows in (a). Biointerfacial water spectra from different subcellular locations: (d) Inside Nucleus regions (total number = 52) (e) Outside Nucleus region (total number = 42). (f) Averaged biointerfacial water spectra of different locations. (g) Positive correlation between the intensities of the two main peaks in biointerfacial water spectra.

Experimental Evidence 3: In-vitro protein hydration shell:

To further substantiate our conclusion, we added Fig. S15, representing in-vitro experiments on the hydration shells of globular proteins. We selected two globular proteins, Bovine Serum Albumin (BSA) and lysozyme, and prepared their aqueous solutions at concentrations ranging from 50 – 150 mg/ml. This concentration range was carefully chosen to prevent protein aggregation. We also obtained dehydrated protein spectra from dry films of these proteins, made through vacuum dehydration over two days. Employing similar Raman-MCR analysis as in our cell studies, we extract the Raman spectra of protein hydration shell at various concentration (Fig. S15). Qualitatively, the hydration shell of BSA and lysozyme measured in-vitro closely mirror the spectral shape of bio-interfacial water observed in single cells (example indicated by the green line). The high resemblance is characterized by two prominent peaks and displayed a more disordered structure ($A_{3290}/A_{3460} < 1$). However, the in-vitro spectra exhibit narrower peak widths compared to cellular spectra, likely due to the averaging effects across diverse proteins within cells, suggesting a more complex interaction landscape in vivo. These findings, while showing some heterogeneity, provide valuable comparative insights and bolster our hypothesis regarding the dominant influence of protein interactions on cellular water spectra.

Fig. S15. *In-vitro* experiments on globular protein hydration shell. (a-d) Raman-MCR analysis on BSA (50 mg/ml in DI water) and Lysozyme (50 -150 mg/ml in DI water) solutions. The BSA and Lysozyme Raman spectra are obtained at 50 mW over 300 s. **(e)** Comparison between the in-vivo bio-interfacial water spectra from single cells and the in-vitro extracted hydration shell spectra of BSA and Lysozyme.

3. Finally, although the biology community may embrace a picture where the majority of water in a cell is not bulklike, this is not a particularly surprising result for researchers with more physical views of aqueous systems. It is a little offensive to insinuate (as the authors do on page 9), that their vibrational spectroscopy results are somehow superior to dynamical measures that have shown a large preponderance of bulklike water in cells. Given the 55 M concentration of bulk water, there is a lot of opportunity for water molecules to find each other to interact with and provide a highly bulklike environment even when there are a lot of other materials in close proximity.

Response: We appreciate the reviewer's perspective and acknowledge the diverse views within the scientific community regarding the nature of water in cellular environments. Our intention was not to undermine the value of dynamical studies but to contribute a *complementary* vibrational spectroscopy viewpoint to the ongoing discourse. We acknowledge the prevalence of bulk-like water in cells, as highlighted by previous research. Our findings aim to enrich the dialogue and propose that macromolecular crowding and bulk-like water properties can coexist, thereby advancing our collective understanding of intracellular water behavior.

That being said, we still would like to point out that the notion of “most water in cytoplasm behaves very differently from bulk water” had historically been very popular and prevailing in the community. We even can see its many followers and supporters in recent literature. As we mentioned in the manuscript, “*In fact, Pollack et al. state in the preface that “practically all cell water is interfacial”²². Recent reviews further argued that, the average distance between macromolecules in the cytoplasm is around 1 nm, corresponding to just three to four molecular layers of water—which, based on classical solvation theory, cannot be considered bulk-like²³⁻²⁵.*” These reviews [ref. 23-25] are rather recent. In light of this, we feel it is still valuable to present both sides of the debate and offer an enriched and more comprehensive picture.

Response to Reviewer 3:

This manuscript presents the adaption of Raman multivariate curve resolution (MCR) spectroscopy or solvation shell spectroscopy to living cells. The authors find that contrary to common belief, living cells consists mostly of bulk water and only about 3% of biointerfacial water with a significantly altered hydrogen-bonding structure. This corresponds to about one water layer around globular proteins inside the cells. This is in contrast to the normal picture of a crowded cell with mostly biointerfacial water. This is a very interesting result and a beautiful illustration of what pushing the limits of solvation shell spectroscopy can do. The study is very well performed with all the necessary control experiments. The manuscript is furthermore well written and appropriate for Nature Communications.

I only have a couple of minor comments for consideration but otherwise the manuscript is essentially publishable as is.

We sincerely thank the reviewer for the support of our work and for the suggestions. We have addressed the individual issues as follows:

Minor comments:

1. In the introduction it says: "Vibrational sum frequency generation (SFG) has intrinsic interface selectivity but requires an extended planar interface^{30,31}". This is strictly speaking not correct. Chirality makes SFG allowed in the bulk and the chirality transfer from biomolecule to the surrounding solvent makes it possible to probe biointerfacial water directly, being the water directly bound to the biomolecule. Actually, later in the manuscript, a chiral SFG study from Elsa Yan and coworkers (reference 70) is described, which illustrates exactly this. However, this study was predated by 2 studies of Petersen and coworkers: <https://doi.org/10.1021/acscentsci.7b00100> and <https://www.science.org/doi/full/10.1126/sciadv.aao5603>. Accordingly, chiral SFG can indeed be used to probe biointerfacial water through this chirality transfer in a label-free way. Even though these examples study biomolecules attached to flat interfaces, chiral SFG does not require a flat surface. However, this chirality transfer is indeed very short range since it relies on the chirality transfer from biomolecule to solvent, as the authors discuss in connection with reference 70.

Response: We are grateful for the reviewer's feedback regarding our discussion on Vibrational SFG spectroscopy. Based on the reviewer's suggestions, we have revised our introduction to highlight the role of chirality transfer in the study of biointerfacial water. Additionally, we have included citations to the two studies recommended by the reviewer, further enriching our manuscript's context and accuracy.

However, living cells present a formidable challenge. Vibrational sum frequency generation (SFG) has intrinsic interface selectivity but requires an extended planar interface³⁰⁻³¹, which is not compatible with intracellular water. **Chiral SFG, leveraging**

chirality transfer from biomolecules to adjacent water, allows probing of interfacial water *in vitro* without the necessity for a flat surface^{32,33}. Vibrational sum frequency scattering can probe the surface of submicron particles in suspension but cannot study biomolecules (such as proteins) that are much smaller than the wavelength of light^{34,35}

On page 5 of the manuscript it reads: "Raman-MCR has been primarily used in small molecules such as alcohols and ions with relatively large hydration shell and hence negligible solute contribution³⁴." This is correct for Raman-MCR, but I would like to point out that the MCR method has previously been applied to biomolecules in the form of FTIR MCR: <https://doi.org/10.1021/acs.jpcllett.6b02925>

Response: We thank reviewer for pointing out the applications of FTIR-MCR to study biomolecules. We have revised manuscript as follows:

"Raman-MCR has been primarily used in small molecules such as alcohols and ions with relatively large hydration shell and hence negligible solute contribution. *In parallel, a related approach has been utilized whereby MCR is combined with FITR spectroscopy to analyze solvation shell of antifreeze proteins in vitro.*"

References:

1. B. Auer and J. Skinner, *The Journal of Chemical Physics* **128** (22) (2008).
2. M. Yang and J. Skinner, *Phys. Chem. Chem. Phys.* **12** (4), 982-991 (2010).
3. M. Ahmed, V. Namboodiri, A. K. Singh, J. A. Mondal and S. K. Sarkar, *J Phys Chem B* **117** (51), 16479-16485 (2013).
4. A. Bandyopadhyay and J. A. Mondal, *Journal of Molecular Liquids* **390**, 122987 (2023).
5. D. Ben-Amotz, *J. Am. Chem. Soc.* **141** (27), 10569-10580 (2019).
6. N. Ghosh, S. Roy, A. Bandyopadhyay and J. A. Mondal, *Liquids* **3** (1), 19-39 (2022).
7. K. R. Fega, A. S. Wilcox and D. Ben-Amotz, *Appl Spectrosc* **66** (3), 282-288 (2012).
8. A. Patra, S. Roy, S. Saha, D. K. Palit and J. A. Mondal, *The Journal of Physical Chemistry C* **124** (5), 3028-3036 (2020).

Reviewers' Comments:

Reviewer #1:

Remarks to the Author:

The authors have carefully considered all the concerns raised by this reviewer and revised/modified accordingly. The manuscript may be published in its present form.

Reviewer #2:

Remarks to the Author:

This review is presented in response to the prompts provided by the journal.

What are the noteworthy results?

This manuscript reports results from innovative experiments that are designed to identify and probe interfacial water in living cells. The authors enlist the Raman MCR method with very high spectral quality spectra from confocal spontaneous Raman microspectroscopy. The result – that most of the intracellular water is bulklike and only a minuscule fraction is not – is appropriate for publication in Nature Communications. It is impressive that the authors can measure Raman spectra from single cells with such high resolution that they can tease out the small interfacial water contribution.

Will the work be of significance to the field and related fields?

Yes, this work should be highly relevant to biological research fields. I believe that the main point the authors wish to communicate is that the vibrational spectrum shows most of the intracellular water appears bulk-like with only a small fraction perturbed by interfaces. They note that the prevailing view of a cell presents water as mostly perturbed by the crowded intracellular environment, but the authors' results refute this.

The paper should generate excitement in the biology community - if they can wade through the dense beginning to get to the punchline. Unfortunately, as written, the paper is unlikely to reach that audience -biologists, biochemists, doctors, etc. – because it is hard to wade through technical details to get to the major result of the work.

How does it compare to the established literature?

The work enlists a published method, Raman MCR spectroscopy and demonstrates its application in confocal Raman microspectroscopy. The application of Raman MCR in microscopy is a tour de force.

Does the work support the conclusions and claims, or is additional evidence needed?

The last section about Thermodynamic Implications includes speculation that is simply not warranted. Nothing about the measurements performed is directly related to thermodynamics. This section is complete speculation and should be removed.

Are there any flaws in the data analysis, interpretation and conclusions?

The interpretation is largely valid. The authors might soften their interpretation of how the vibrational spectral signature relates to water dynamics. The authors could acknowledge the large amount of work that has demonstrated facets of interfacial water, both structure and dynamics. Quite a few publications report vibrational spectroscopy studies of water in confined environments where perturbation appears in the first water layer near interfaces. However, dynamics may not follow the same trends (see Piletic et al. JPCA 2006, Garret and Baiz, JPCB 2023).

Do these prohibit publication - NO - or require revision -YES?

Yes, the Thermodynamics section at the end is entirely speculative and should be removed from the paper.

The authors also make statements in the paper that are simply wrong and should be removed from the paper. For example,

--SFG measurements have been successfully performed on droplets so they do not "...require a planar interface..." as stated in the second paragraph of the introduction. Roke and coworkers have demonstrated in several publications that it is possible to obtain SFG data from submicron diameter droplets.

--In the first paragraph of the Results section, the authors state, "the broad high-wavenumber region from 3100 to 3800 cm^{-1} is mostly from water" which true. But they go on to write, "...hence often called the O-H stretching region." The region is not called the O-H stretching region because it is from water. It is called the O-H stretching region because O-H stretching vibrations of any molecule appear in this spectral range.

Although the results are very interesting, the presentation of the work leaves a lot to be desired and could be significantly improved for better readability. The authors provide a tremendous amount of detail about intricate details of the experiments and analysis. The figures are hard to understand even with careful reading of the paper's text and certainly are not clear from the figure captions. Most of the figures could be simplified by moving some panels into the SI including Figure 1a (and accompanying text), Figures 3 g, h, i, and j. Most of the panels of Figure 2 verifying the authors' ability to measure -OH rather than -NH signals at frequencies $>3000 \text{ cm}^{-1}$. Any place where the authors require readers spend significant effort in the SI should probably just appear in the SI, not the main paper.

Is the methodology sound? Does the work meet the expected standards in your field?

Yes. The revised SI allows the reader to understand how the authors applied the MCR method.

Is there enough detail provided in the methods for the work to be reproduced?

Yes, although virtually no one else can do this experiment because it is so highly specialized.

Minor issues:

On page 7, the first line of the highlighted paragraph includes the word "interpretating" which should be "interpreting".

Throughout the paper the authors misuse the word "spectra", which is the plural of "spectrum". They need ensure that they use the singular and plural versions appropriately. In many of the figure caption, the word "spectra" should be "spectrum". For example, the legend in Figure 1b should read "Single Cell Spectrum A" or better simply "Single Cell A".

Reviewer #3:

Remarks to the Author:

I am satisfied with the authors changes to the manuscript. I will now recommend the manuscript for publication.

Response to Reviewer 1:

The authors have carefully considered all the concerns raised by this reviewer and revised/modified accordingly. The manuscript may be published in its present form.

Response: We thank reviewer 1 for their constructive feedback on our manuscript. We are glad to see that the manuscript is now suitable for publication in its present form.

Response to Reviewer 2:

What are the noteworthy results?

This manuscript reports results from innovative experiments that are designed to identify and probe interfacial water in living cells. The authors enlist the Raman MCR method with very high spectral quality spectra from confocal spontaneous Raman microspectroscopy. The result – that most of the intracellular water is bulklike and only a minuscule fraction is not - is appropriate for publication in Nature Communications. It is impressive that the authors can measure Raman spectra from single cells with such high resolution that they can tease out the small interfacial water contribution.

Response: We appreciate reviewer's positive evaluation of our manuscript. We are pleased to see the innovation of our experiments has been recognized.

Will the work be of significance to the field and related fields?

Yes, this work should be highly relevant to biological research fields. I believe that the main point the authors wish to communicate is that the vibrational spectrum shows most of the intracellular water appears bulk-like with only a small fraction perturbed by interfaces. They note that the prevailing view of a cell presents water as mostly perturbed by the crowded intracellular environment, but the authors' results refute this.

The paper should generate excitement in the biology community - if they can wade through the dense beginning to get to the punchline. Unfortunately, as written, the paper is unlikely to reach that audience -biologists, biochemists, doctors, etc. – because it is hard to wade through technical details to get to the major result of the work.

Response: We appreciate reviewer 's recognition of the significance of our work to the biological research community. To address reviewer's concern, we have revised the manuscript to clarify the main results and reduce the density of technical details, ensuring that the key findings are more easily understood by a diverse readership.

How does it compare to the established literature?

The work enlists a published method, Raman MCR spectroscopy and demonstrates its application in confocal Raman microspectroscopy. The application of Raman MCR in microscopy is a tour de force.

Response: We appreciate reviewer's recognition of our novel application of Raman MCR spectroscopy.

Does the work support the conclusions and claims, or is additional evidence needed?

The last section about Thermodynamic Implications includes speculation that is simply not warranted. Nothing about the measurements performed is directly related to thermodynamics. This section is complete speculation and should be removed.

Response: We appreciate reviewer's critical insights on the section about Thermodynamic Implications. We have deleted this section to ensure that all conclusions and claims are firmly supported by the provided evidence.

Are there any flaws in the data analysis, interpretation and conclusions?

The interpretation is largely valid. The authors might soften their interpretation of how the vibrational spectral signature relates to water dynamics. The authors could acknowledge the large amount of work that has demonstrated facets of interfacial water, both structure and dynamics. Quite a few publications report vibrational spectroscopy studies of water in confined environments where perturbation appears in the first water layer near interfaces. However, dynamics may not follow the same trends (see Piletic et al. JPCA 2006, Garret and Baiz, JPCB 2023).

Response: We appreciate reviewer's assessment of our data analysis and interpretation. We have removed the section on water dynamics implications to ensure that all conclusions and claims are firmly supported by the provided evidence.

Do these prohibit publication - NO - or require revision -YES?

Yes, the Thermodynamics section at the end is entirely speculative and should be removed from the paper.

Response: We have removed the speculative Thermodynamics section from the paper.

The authors also make statements in the paper that are simply wrong and should be removed from the paper. For example, --SFG measurements have been successfully performed on droplets so they do not "...require a planar interface..." as stated in the second paragraph of the introduction. Roke and coworkers have demonstrated in several publications that it is possible to obtain SFG data from submicron diameter droplets.

Response: We appreciate the reviewer's comments. We have indeed referenced the work of Roke and colleagues in *references 34 and 35*, where the successful application of SFG on submicron diameter droplets is demonstrated. This inclusion was part of our last revision, addressing the

exact concern about the need for a planar interface in traditional SFG methods. We regret any oversight if this adjustment was not identified.

“Vibrational sum frequency generation (SFG) has intrinsic interface selectivity but requires an extended planar interface^{30,31}, which is not compatible with intracellular water. Chiral SFG, leveraging chirality transfer from biomolecules to adjacent water, allows probing of interfacial water *in vitro* without the necessity for a flat surface^{32,33}. **Vibrational sum frequency scattering can probe the surface of submicron particles in suspension** but cannot study biomolecules (such as proteins) that are much smaller than the wavelength of light^{34,35}”

--In the first paragraph of the Results section, the authors state, “the broad high-wavenumber region from 3100 to 3800 cm⁻¹ is mostly from water” which true. But they go on to write, “...hence often called the O-H stretching region.” The region is not called the O-H stretching region because it is from water. It is called the O-H stretching region because O-H stretching vibrations of any molecule appear in this spectral range.

Response: We have revised the wording to eliminate any confusion in logic. The text now states: **“the broad high-wavenumber region from 3100 to 3800 cm⁻¹, referring to the O-H stretching region, is mostly from water”**. This change ensures clarity and corrects the earlier misinterpretation.

Although the results are very interesting, the presentation of the work leaves a lot to be desired and could be significantly improved for better readability. The authors provide a tremendous amount of detail about intricate details of the experiments and analysis. The figures are hard to understand even with careful reading of the paper's text and certainly are not clear from the figure captions. Most of the figures could be simplified by moving some panels into the SI including Figure 1a (and accompanying text), Figures 3 g, h, i, and j. Most of the panels of Figure 2 verifying the authors' ability to measure -OH rather than -NH signals at frequencies >3000 cm⁻¹. Any place where the authors require readers spend significant effort in the SI should probably just appear in the SI, not the main paper.

Response: We appreciate the reviewer's suggestions to enhance readability by relocating some figure panels to the Supplementary Information (SI). However, after thorough consideration, we have chosen to keep these figures in the main paper. Our decision is driven by the belief that these figures (Fig. 1a, most panels of Fig. 2, and Fig. 3g, h, i, j) are crucial to the core narrative and understanding of our manuscript. Moving them to the SI could interrupt the flow and possibly obscure key insights for readers who may not consult the SI. Specifically, for Fig. 3g, h, i, and j, their purpose is to demonstrate the robustness of our methodology and to illustrate both the general similarity and subtle differences among cells. Presenting these together in the main text allows these nuances to be more clearly observed. We believe that retaining these figures in the main text ensures that all readers have immediate access to essential data that supports our conclusions.

Is the methodology sound? Does the work meet the expected standards in your field?
Yes. The revised SI allows the reader to understand how the authors applied the MCR method.

Response: We appreciate the reviewer's acknowledgment that our methodology is sound and that the revisions to the Supplementary Information (SI) have clarified our application of the MCR method.

Is there enough detail provided in the methods for the work to be reproduced?

Yes, although virtually no one else can do this experiment because it is so highly specialized.

Response: We appreciate the reviewer's recognition that our methods are detailed enough for reproducibility. While our experiment is specialized, we would like to emphasize that the results are indeed replicable by others with access to confocal Raman micro-spectroscopy.

Minor issues:

On page 7, the first line of the highlighted paragraph includes the word “interpretating” which should be “interpreting”.

Throughout the paper the authors misuse the word “spectra”, which is the plural of “spectrum”. They need ensure that they use the singular and plural versions appropriately. In many of the figure caption, the word “spectra” should be “spectrum”. For example, the legend in Figure 1b should read “Single Cell Spectrum A” or better simply “Single Cell A”

Response: We appreciate the reviewer's attention to detail in identifying these linguistic errors. We have corrected "interpretating" to "interpreting" on page 7 and have thoroughly reviewed the use of "spectra" and "spectrum" throughout the paper, ensuring the correct singular and plural forms are used, including in the figure captions as specified.

Response to Reviewer 3:

I am satisfied with the authors changes to the manuscript. I will now recommend the manuscript for publication.

Response: We appreciate the reviewer's recommendation for the publication of our manuscript.